# Implicit Safety Alignment from Crowd Preferences

Qian Lin [1]   Daniel S. Brown [1]

## Abstract

Reinforcement Learning from Human Feedback (RLHF) can reveal implicit objectives such as safety considerations that go beyond task completion. In this work, we focus on the common safety criteria embedded in crowd preference datasets, where different users may express distinct preferences or objectives, yet follow similar safety principles. Our aim is to discover shared safety criteria from crowd preferences and then transfer them to downstream RL tasks to regularize agent behavior and enforce safety. We first show that direct reward combination—optimizing a preference-learned reward model together with downstream task rewards—has inherent limitations. Motivated by this, we propose Safe Crowd Preference-based RL, a hierarchical framework that extracts safety-aligned skills from crowd preferences and composes them via a high-level policy to safely solve downstream tasks. Experiments across safe RL environments and a preliminary LLM-style task with diverse user goals and shared safety constraints demonstrate that our approach substantially lowers safety costs without access to explicit safety rewards, while achieving task performance comparable to oracle methods trained with ground-truth safety signals.

## 1. Introduction

Reinforcement learning (RL) (Sutton et al., 1998) with well-designed rewards that encode task-specific information enables agents to accomplish their goals. However, specifying reward functions that fully capture desired behavior is notoriously difficult, even for seemingly simple tasks, and imperfect or misspecified rewards are common in practice (Booth et al., 2023; Malik et al., 2021). In complex decision-making scenarios, reward designers often focus on task completion while inadvertently omitting additional objectives that are ambiguous, implicit, or hard to formalize—particularly safety-related considerations. For instance, in driving-related tasks, one can assign a sparse reward for simply reaching the destination, yet real-world driving also involves implicit safety considerations—such as maintaining safety margins and discouraging aggressive or risky behaviors—that are difficult to encode accurately through manual reward design. Optimizing such incomplete rewards can therefore lead to hazardous behavior, highlighting the limitations of purely hand-specified reward functions.

Reinforcement Learning from Human Feedback (RLHF) has been developed to overcome the limitations of human-designed rewards and has been widely applied in large language models (LLMs) (Ouyang et al., 2022; Bai et al., 2022), robotics (Hejna III & Sadigh, 2023; Liu et al., 2023a), and games (Christiano et al., 2017; Brown et al., 2019). It enables the incorporation of implicit objective information into decision-making by learning new reward functions from scratch (Christiano et al., 2017), discovering additional objectives (Dai et al., 2024), and directly modeling expected behaviors (An et al., 2023; Hejna et al., 2024) to align policies with human preferences. In practice, preference data are often collected from a crowd of users. While individual users may vary in their task objectives, preferences, or behavioral patterns, they often share implicit criteria—such as avoiding unsafe states and actions. This shared structure suggests the possibility of isolating latent behavioral criteria that are consistent across crowd preferences and reusing them across downstream tasks, thereby reducing the need for additional human demonstrations or preference labels.

Accordingly, we focus on safety-critical scenarios under crowd preference learning, where safety-agnostic task rewards inadvertently promote unsafe behaviors, while crowd preferences—collected across users with diverse goals yet shared safety criteria—provide a complementary safety signal to compensate for such deficiencies. Our goal is to discover such reusable, implicit safety-related criteria embedded in crowd-user, cross-task preferences and transfer them to downstream tasks, where only safety-agnostic task rewards are available, to learn safe policies. A natural approach is to learn a reward model from crowd preferences via standard RLHF and then optimize a combination of this preference-learned reward and the downstream task

---

[1]Kahlert School of Computing, University of Utah. Correspondence to: Qian Lin <qian.lin@utah.edu>.

*Proceedings of the 43rd International Conference on Machine Learning*, Seoul, South Korea. PMLR 306, 2026. Copyright 2026 by the author(s).

reward. However, our theoretical and empirical analyses reveal two limitations of this reward combination approach: (1) the learned reward model may capture not only the intended shared criteria but also user-specific components from crowd preferences, which can degrade downstream performance—especially when the dataset is imbalanced across users; and (2) the performance is sensitive to the weighting between the preference-learned and task rewards, making the method difficult to tune.

To overcome these limitations, we propose Safe Crowd Preference-based Reinforcement Learning, a hierarchical framework that composes policies rather than rewards. Specifically, we first model diverse user-preferred behaviors as low-level skills, learned from crowd preference data via a VAE-based latent skill model and existing RLHF techniques. For a new task, we then train a high-level policy to compose these skills by maximizing only the downstream task reward. Since all skills are aligned with crowd preferences and thus inherently encode shared safety criteria, composing them to solve new tasks naturally yields safe behaviors without requiring explicit safety signals. We evaluate our approach on a suite of safe RL environments and a proof-of-concept LLM-style task with diverse task goals but shared safety constraints. Empirically, directly optimizing task rewards leads to severe constraint violations, whereas our method significantly reduces safety costs without requiring explicit safety rewards and achieves task performance comparable to oracle baselines trained with ground-truth safety signals. We further find that reward combination (i.e., directly optimizing a weighted sum of the preference-learned reward and the new task reward) is highly sensitive to preference imbalance: performance drops sharply when one annotator contributes the majority of the data, while our method is significantly more robust to such imbalance.

## 2. Related Work

### 2.1. Reinforcement Learning from Human Feedback (RLHF)

Reinforcement Learning from Human Feedback (RLHF) aims to improve decision making by aligning agents with various forms of human feedback, including preferences (Wirth et al., 2017), demonstrations (Ng et al., 2000), trajectory rankings (Brown et al., 2019; 2020), corrections (Bajcsy et al., 2017), and terminations (Hadfield-Menell et al., 2017). This work is most closely related to Preference-based RL (PbRL), where binary comparisons over trajectories are used to guide policy learning. Classical PbRL methods (Lee et al., 2021; Christiano et al., 2017) typically learn a reward model using the Bradley–Terry–Luce (BTL) model (Bradley & Terry, 1952), and then perform online RL or offline RL (Shin et al.) based on the learned reward. Recently, several works (An et al., 2023; Hejna et al., 2024) have proposed directly learning a policy un-

der the BTL model without an explicit reward learning stage, thereby avoiding optimization challenges inherent in RL. Both reward-based and direct preference learning approaches are compatible with our framework, and we evaluate both with online and offline RL in our experiments.

### 2.2. Crowd Preference-based RL

Most existing RLHF methods assume that all comparisons are generated by users who share the same set of values and goals. However, in practice, human feedback often reflects diverse trade-offs among multiple objectives or even different decision goals, referred to as crowd preference data. Several studies on crowd preference learning have been proposed to promote fair aggregation of feedback from diverse sources (Chhan et al., 2024; Chakraborty et al., 2024), improve the robustness of reward models against user noise or jailbreaks (Siththaranjan et al., 2024; Boldi et al., 2024), and enable personalized decision-making for individual users (Jang et al., 2023; Poddar et al., 2024). Among them, Poddar et al. (2024) is most closely related to our work, as it also uses a VAE-based architecture to model behavioral diversity implied by crowd preferences. However, it focuses on identifying existing preferences to enhance decision diversity, while our work emphasizes behavioral generalization—combining diverse behaviors to solve new tasks. Furthermore, to our knowledge, we are the first to study shared objectives in crowd preferences and to explore how shared safety objectives can be learned and then used to guide safe downstream policy learning.

### 2.3. Constraint Learning from Human Feedback

In the field of constraint learning, many studies have explored improving policy behavior through human feedback, since optimizing task rewards alone can lead to safety risks, necessitating the additional learning of constraint or penalty rewards. One research direction is inferring constraints from demonstrations, known as Inverse Constraint Reinforcement Learning (ICRL) (Malik et al., 2021; Papadimitriou et al., 2022; Kim et al., 2023). A related approach (Papadimitriou & Brown, 2024) extends this idea by using Bayesian inference to infer constraints from preferences over demonstrations, rather than from demonstrations directly. Another line of work, Safe RLHF (Dai et al., 2024; Chittepu et al., 2025), proposes jointly learning both reward and constraint functions from preference data. However, both ICRL and Safe RLHF require an explicit decomposition between task and safety objectives, either by assuming access to the task reward used to generate demonstrations or by representing different objectives with distinct types of preference labels. In contrast, we adopt a more realistic setting in which preferences reflect an unknown mixture of multiple underlying objectives, without prior knowledge of these objectives.

## 2.4. Skill Discovery

Skill discovery has been widely studied as a means to address long-horizon challenges (Pertsch et al., 2021; Lioutikov et al., 2017), improve exploration and data efficiency (Nachum et al., 2018; 2019; Eysenbach et al., 2018), and facilitate offline RL (Ajay et al., 2020). Most existing approaches are unsupervised, extracting skills from demonstrations or replay buffers using encoder–decoder architectures (e.g., VAEs) (Shankar & Gupta, 2020; Ajay et al., 2020) or information-theoretic objectives (Eysenbach et al., 2018). The resulting skills are applied to downstream tasks via hierarchical policy learning (Bacon et al., 2017; Nachum et al., 2018) or as behavioral priors constraining policy optimization (Singh et al., 2020). Our work departs from this line of research in two key aspects. First, rather than modeling behavioral diversity from demonstrations, we consider both preference diversity and commonality and learn skills from crowd preference data. Second, beyond improving downstream task performance, we also aim to preserve the shared criteria encoded in crowd preferences. A closely related work is skill preference learning (Wang et al., 2022), which incorporates preferences to specify downstream objectives but still relies on demonstration-based skill extraction.

## 3. Preliminaries

General reinforcement learning from human feedback (RLHF) can be formulated as a reward-free Markov decision process (MDP) $\mathcal{M}/r = (\mathcal{S}, \mathcal{A}, \mathcal{P}, \gamma)$, where $\mathcal{S}$ is the state space, $\mathcal{A}$ is the action space, $\mathcal{P}(s_{t+1} \mid s_t, a_t)$ is the transition dynamics, and $\gamma$ is the discount factor. The true reward $r(s, a)$ is unobservable but implicitly reflected in the preference dataset $\mathcal{D}_{\text{pref}} = \{(\tau^1, \tau^2, y)_i\}_{i=1}^N$, where each trajectory segment $\tau = (s_1, a_1, s_2, a_2, \ldots, s_L, a_L)$ has length $L$. The probability of a preference label $y$ is given by the Bradley–Terry model:

$$P(y = 1|\tau^1, \tau^2) = 1 - P(y = 0|\tau^1, \tau^2)$$
$$= \frac{\exp \beta \cdot u(\tau^1)}{\exp \beta \cdot u(\tau^1) + \exp \beta \cdot u(\tau^2)}, \quad (1)$$

where $y = 1$ indicates that $\tau^1$ is preferred over $\tau^2$ and $y = 0$ otherwise, $u$ is the trajectory utility, and $\beta$ models human rationality. In this work, we consider two mainstream preference models: the partial-return model (Christiano et al., 2017) and the regret-based model (Knox et al., 2022; Hejna et al., 2024). These two models correspond to different utilities, $u$, and lead to distinct solution formulations.

**Partial-return model.** This model adopts the cumulative reward $u(\tau^i) = \sum_{t=1}^L r(s_t^i, a_t^i)$ as the utility to generate the preference labels. In this case, a typical solution is to first learn a reward function $r_\phi(s, a)$ using a maximum

likelihood estimate, where the likelihood is given by

$$P(y = 1|\tau^1, \tau^2) = \frac{\exp \hat{u}(\tau^1)}{\exp \hat{u}(\tau^1) + \exp \hat{u}(\tau^2)}, \quad (2)$$

where $\hat{u}(\tau^i) = \sum_{t=1}^L r_\phi(s_t^i, a_t^i)$. A policy, $\pi_\phi$, is then obtained by optimizing the learned reward using RL.

**Regret-based model.** This model uses the advantage function of the optimal policy $\pi^*$ under the true reward $r$, i.e., $u(\tau^i) = \sum_{t=1}^L A^*(s_t^i, a_t^i) = \sum_{t=1}^L Q^*(s_t^i, a_t^i) - V^*(s_t^i)$, as the utility to generate the preference labels. To recover $\pi^*$, one can adopt the solution induced by the partial-return preference model: although the learned reward effectively approximates the optimal advantage function rather than the true reward, maximizing it still induces an equivalent optimal policy (Knox et al., 2024; Tien & Brown, 2023). In this work, we consider an alternative, Contrastive Preference Learning (CPL), since it enables a purely supervised learning approach that avoids the challenges of RL optimization (Hejna et al., 2024). Specifically, CPL leverages the relationship between the optimal advantage function $A^*$ and the optimal policy $\pi^*$ to derive an instantiation of the Bradley–Terry model that only depends on the policy. Instead of learning a reward, CPL directly learns a policy $\pi_\theta$ by maximizing the likelihood of human preferences:

$$P(y = 1|\tau^1, \tau^2) = \frac{\exp(\log \pi_\theta(\tau^1))}{\exp(\log \pi_\theta(\tau^1)) + \exp(\lambda \log \pi_\theta(\tau^2))}, \quad (3)$$

where $\log \pi_\theta(\tau^i) = \sum_{t=1}^L \gamma^t \alpha \log \pi_\theta(s_t^i, a_t^i)$, and $\alpha$ is a temperature parameter, and $\lambda$ is a bias regularizer in CPL.

## 4. Limitations of Vanilla RLHF in Crowd Preference Learning

In this section we discuss crowd-RLHF, introduce our novel safety transfer problem under a shared-structure assumption on crowd preferences, and analyze the limitations of Vanilla RLHF as a solution.

### 4.1. Problem Setting

Similar to Siththaranjan et al. (2024) and Poddar et al. (2024), we study the problem of crowd preference learning, where the collected preference data are generated not from a single utility $u(s, a)$, but from multiple utilities $u(s, a, z)$ determined by a hidden context $z$. Therefore, the crowd preference dataset can be denoted as $\mathcal{D}_{\text{pref}} = \{S_z = \{(\tau^1, \tau^2, y_z)_i\}_{i=1}^S \mid z \sim p(z)\}$, where each $S_z$ represents a set of preference pairs provided by a user with hidden context $z$, and $p(z)$ denotes the (unobserved) distribution over contexts. All preference labels $y_z$ within the same set $S_z$ are assumed to be generated using the same utility $u(\cdot, z)$. In practice, however, the true context $z$ associated with each $S_z$ and its corresponding utility $u(\cdot, z)$ are not observed.

In contrast to prior work, we consider the fact that, while different users may evaluate trajectories with diverse human values, they will likely still share common decision criteria, such as avoiding unsafe behaviors. Formally, we assume that the rewards underlying the different crowd members' utilities can be decomposed as $r(s, a, z) = r_{\text{user}}(s, a, z) + r_{\text{share}}(s, a)$. Here, $r_{\text{user}}(s, a, z)$ captures crowd variations, modeling differences in task objectives or goals across users, while $r_{\text{share}}(s, a)$ represents criteria that are shared among all users. Although the decomposition into task-specific and shared components is not unique, in this work we focus on safety as the shared criterion. Specifically, users are assumed to share state–action safety constraints, which are modeled via a shared reward component:

$$r_{\text{share}}(s, a) = \begin{cases} -K, & \text{if } (s, a) \in X_{\text{unsafe}}, \\ 0, & \text{otherwise,} \end{cases} \quad (4)$$

where $K > 0$ is a large constant (ideally $K \to \infty$ to completely prohibit unsafe state–action pairs), and $X_{\text{unsafe}}$ denotes the set of unsafe state–action pairs. [1]

Our goal is not only to discover such shared safety criteria from crowd preference data, but also to transfer them to new tasks. Concretely, we consider a downstream task with a safety-agnostic reward $r_{\text{new}}(s, a)$, which is imperfect in that it encourages task completion but may induce unsafe behaviors. Such imperfect or misspecified rewards are common in practice, as reward design is inherently difficult and often fails to capture implicit objectives (Krakovna et al., 2020; Booth et al., 2023; Tien & Brown, 2023). Thus, the ideal downstream policy should optimize

$$\pi^* = \arg\max_{\pi} \mathbb{E}\Big[ \sum_t r_{\text{new}}(s_t, a_t) + r_{\text{share}}(s_t, a_t) \Big].$$

even though the true shared reward $r_{\text{share}}$ is not directly observable in practice.

### 4.2. Does Vanilla RLHF Work?

One intuitive approach is to directly learn a single reward model $\hat{r}(s, a)$ using vanilla reward learning (Eq. 2) on crowd preferences to capture shared user criteria, which induces a trajectory-level utility $\hat{u} = \sum_t \hat{r}(s_t, a_t)$, and then combine it with the new task reward:

$$r'_{\text{new}}(s, a) = (1 - \omega) r_{\text{new}}(s, a) + \omega \hat{r}(s, a), \quad (5)$$

where $\omega$ is a trade-off weight. To analyze the properties of $\hat{r}$, we consider preferences over complete trajectories and introduce the following definition:

**Definition 4.1.** A trajectory pair $(\tau^1, \tau^2)$ is *consistent* if

$$u(\tau^1, z) \geq u(\tau^2, z) \text{ for all } z \text{ or } u(\tau^1, z) \leq u(\tau^2, z) \text{ for all } z.$$

[1]Extensions to more general forms of safety consensus in crowd preferences are discussed in Appendix F.

Otherwise, the pair is called *inconsistent*.

That is, all users agree on the relative preference ordering between $\tau^1$ and $\tau^2$. Denote safe trajectories as those satisfying $\forall (s, a) \in \tau^{\text{safe}}, (s, a) \notin X_{\text{unsafe}}$, and unsafe trajectories otherwise. We then have the following result:

**Theorem 4.2.** *Let the true utility be* $u(\tau, z) = \sum_t r_{\text{user}}(s_t, a_t, z) + r_{\text{share}}(s_t, a_t)$, *where the shared component* $r_{\text{share}}$ *represents a safety penalty:*

$$r_{\text{share}}(s, a) = \begin{cases} -K, & (s, a) \in X_{\text{unsafe}}, \\ 0, & \text{otherwise.} \end{cases}$$

*If the penalty* $K > 2L \max_{s,a,z} |r_{\text{user}}(s, a, z)|$, *then all pairs* $(\tau^{\text{safe}}, \tau^{\text{unsafe}})$ *consisting of one safe and one unsafe trajectory are consistent. Consequently, the single reward model* $\hat{u}$ *learned from infinite crowd preference data satisfies*

$$\hat{u}(\tau^{\text{safe}}) > \hat{u}(\tau^{\text{unsafe}}),$$

*for any* $\tau^{\text{safe}}$ *and any* $\tau^{\text{unsafe}}$.

See proof in Appendix A.1. This is an intuitive conclusion: if all users consistently prefer safe trajectories over unsafe ones, the learned reward model will also prefer safe ones.

Although we have shown that a single reward model can capture the safety consensus, this approach suffers from two critical issues: 1) It requires carefully fine-tuning the trade-off weight $\omega$ between $r_{\text{new}}(s, a)$ and $\hat{r}(s, a)$, especially when their numerical scales differ significantly; 2) The learned model $\hat{r}(s, a)$ may encode not only the shared safety preference but also user-specific components, which can mismatch with $r_{\text{new}}(s, a)$ in downstream training. The following theorem formalizes this situation, showing that the issue becomes more severe when the number of preferences from different hidden contexts $z$ is imbalanced.

**Theorem 4.3.** *Assume a finite set of hidden contexts* $\{z_1, \dots, z_m\}$ *and a finite trajectory set* $\mathcal{T}$. *Denote* $X_{\text{ics}}$ *as the set of all inconsistent pairs, define*

$$N(\tau^1, \tau^2, z) = \sum_{\tau \in \mathcal{T}} \big( \mathbb{I}\{u(\tau^1, z) > u(\tau, z) > u(\tau^2, z)\} + \mathbb{I}\{u(\tau^1, z) < u(\tau, z) < u(\tau^2, z)\} \big).$$

*For any* $z_k$, *if the probability of* $z_k$ *satisfies*

$$p(z_k) > \frac{|\mathcal{T}| - 1}{\min_{(\tau, \tau') \in X_{\text{ics}}} N(\tau, \tau', z_k) + |\mathcal{T}|}$$

*then for all* $\tau^1, \tau^2 \in \mathcal{T}$,

$$\hat{u}(\tau^1) > \hat{u}(\tau^2) \Longleftrightarrow u(\tau^1, z_k) > u(\tau^2, z_k).$$

The proof is provided in Appendix A.1. This result provides an upper bound: once the proportion of preference data

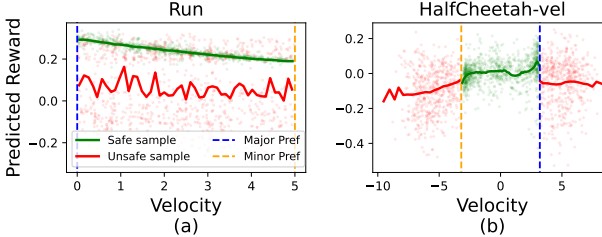

*Figure 1.* Predicted rewards on safe and unsafe samples under the *imbalanced* crowd-preference setting. Safe trajectories (green) receive higher rewards than unsafe ones (red), while trajectories aligned with the majority preference (blue) are over-rewarded compared to minority-preferred ones (yellow).

from a task $z_k$ exceeds this threshold, the learned model's preference ordering aligns completely with $u(\tau, z_k)$. Consequently, maximizing the learned utility yields trajectories that are optimal for task $z_k$, i.e., $\arg\max_\tau \hat{u}(\tau) = \arg\max_\tau u(\tau, z_k)$. However, since the optimal trajectories under different context $z$ can be potentially disjoint, the learned reward will bias policy optimization toward the optimal trajectories under $z_k$, deviating from those optimal under $r_{\text{new}}$. This mismatch in task rewards directly harms the performance on the target task.

We empirically validate these conclusions by training a reward model on an imbalanced crowd preference dataset, where the majority of users favor a specific target and contribute ten times more data than the others. In Run, the majority prefers a lower target velocity (0) over a higher velocity (5), while all trajectories must avoid constrained regions; in HalfCheetah, the majority prefers positive over negative velocity, subject to velocity limits (see Appendix C for details). Figure 1 shows that the learned reward assigns higher values to safe trajectories than unsafe ones, consistent with Theorem 4.2, but also over-rewards trajectories aligned with the majority preference, confirming Theorem 4.3. We further investigate the downstream performance of reward combination (i.e., Eq. 5) under different trade-off weights $\omega$ in Section 6.2, and show that it suffers from strong sensitivity to the trade-off weight as well as performance degradation induced by preference imbalance, as predicted by the Theorem 4.3.

## 5. Safe Crowd Preference-based Reinforcement Learning

### 5.1. Policy Composition Rather Than Reward Combination

Since it is difficult to disentangle the user-specific component and the shared criteria, it may not be effective to directly incorporate the learned reward into downstream optimization as in Eq. 5. Instead, our idea is to optimize only the downstream reward $r_{\text{new}}$ while restricting the policy search to behaviors consistent with the crowd preferences. Specifically, we constrain the agent to compose

behaviors drawn from the preference-aligned skill space learned from the crowd preferences. This approach is motivated by two insights: **(1)** constraining policy optimization to preference-aligned behaviors—rather than the raw action space—naturally enforces the shared criteria (e.g., avoiding unsafe state–action pairs in this work); and **(2)** behaviors from users with diverse task objectives provide useful primitives that facilitate downstream task learning. Formally, inspired by hierarchical learning, we treat preference-aligned policies $\pi_l(a|s, z)$ as low-level skills and learn a high-level policy $\pi_h(z|s)$ that composes these skills to solve the downstream task:

$$\max_{\tau \sim \pi_h \cdot \pi_l} \mathbb{E}\Big[\sum_t r_{\text{new}}(s_t, a_t)\Big]. \tag{6}$$

### 5.2. Skill Discovery from Crowd Preference

There exist several RLHF techniques that can learn preference-aligned policies, as discussed in Section 3. However, without access to the true label $z$ for each preference pair, we cannot directly model the behaviors of different users in crowd preference data. Inspired by Variational Preference Learning (VPL) (Poddar et al., 2024), we address this challenge by inferring a latent variable $z'$ from crowd preference data using a Variational Auto-Encoder (VAE), where $z'$ serves as a proxy for the underlying (unobserved) user context $z$ that governs preference behavior. Here, $z$ denotes the true but unobserved context, while $z'$ is a learned latent variable that approximates this context and is used to condition both the reward model and the policy. Specifically, we first learn a latent-conditioned reward:

$$\mathbb{E}_{S_z \sim \mathcal{D}_{\text{pref}}}\Big[\mathbb{E}_{z' \sim q_\psi(z'|S_z)}\Big[\sum_{(\tau^1, \tau^2, y_z) \in S_z} \log P(y|\tau^1, \tau^2, z')\Big] \\ - D_{KL}(q_\psi(z'|S_z)||p(z'))\Big] \tag{7}$$

where $q_\psi$ is an encoder to predict the latent variable $z'$ based on a set of preferences from some user with hidden context $z$ and $p(z')$ is a prior distribution. $P(y|\tau^1, \tau^2, z')$ follows the definition of Eq. 2 except that $r_\phi(s, a)$ is replaced with $r_\phi(s, a, z')$ as a decoder to predict the preference relationship under $z'$:

$$P(y = 1|\tau^1, \tau^2, z') = \frac{\exp \hat{u}(\tau^1, z')}{\exp \hat{u}(\tau^1, z') + \exp \hat{u}(\tau^2, z')}, \tag{8}$$

where $\hat{u}(\tau^i, z') = \sum_{t=1}^k r_\phi(s_t^i, a_t^i, z')$. Then a latent-conditioned $\pi_\theta(a|s, z')$ as our low-level policy can be learned via offline RL methods (e.g., IQL (Kostrikov et al., 2022) in the implementation): for each $z' \sim p(z')$,

$$\max_{\pi_\theta(a|s, z')} \mathbb{E}_{\tau \sim \mathcal{D}_\tau}\Big[\sum_t r_\phi(s_t, a_t, z')\Big], \tag{9}$$

where $\mathcal{D}_\tau$ is an offline trajectory dataset.

However, this approach requires access to an additional offline trajectory dataset $\mathcal{D}_\tau$ and is susceptible to RL optimization instability (Hejna et al., 2024). Therefore, we also propose an alternative approach that incorporates Contrastive Preference Learning (CPL) into the VPL framework under the regret-based preference setting, i.e., by defining $P(y|\tau^1, \tau^2, z')$ in Eq. 7 as

$$P(y = 1|\tau^1, \tau^2, z') = \frac{\exp f(\tau^1|z')}{\exp f(\tau^1|z') + \exp \lambda f(\tau^2|z')},$$
(10)

where $f(\tau^i|z') = \sum_{t=1}^{k} \gamma^t \alpha \log \pi_\theta(a_t^i|s_t^i, z')$. We empirically validate VPL and its novel CPL variant in Section 6 using crowd preferences under the partial-return and regret-based models, respectively.

### 5.3. Downstream Learning

For a new downstream task, instead of predicting actions directly from states, the high-level policy outputs a latent variable that indexes a preference-aligned low-level behavior to generate the actions, i.e., $a \sim \pi_l(a \mid s, z' = \pi_h(s))$.[2] This skill-based action selection keeps the agent within the distribution of preference-aligned behaviors, thereby inheriting the safety preferences encoded in crowd data and avoiding unsafe actions and states. Moreover, given that the VAE objective in Eq. 7 encourages preference-aligned behaviors to be encoded near the prior $p(z')$, we regularize the high-level policy via $L_{\text{reg}} = \log p(z' = \pi_h(s))$, which keeps the latent skill $z'$ selected by $\pi_h$ close to the latent structure learned from crowd preferences and prevents out-of-distribution skills that could lead to unsafe behavior. Following this design, we adopt the TD3 (Fujimoto et al., 2018) framework for high-level policy training using the loss:

$$L_{\pi_h} = -\mathbb{E}_{a \sim \pi_h \cdot \pi_l}\big[Q(s, a) + \beta_{\text{reg}} L_{\text{reg}}\big], \quad (11)$$

where $\beta_{\text{reg}}$ weights the regularization term, $Q$ is the estimated value for $r_{\text{new}}$, and the low-level policy $\pi_l$ is frozen so that gradients propagate through $Q$ and $\pi_l$ into $\pi_h$.

Eq. (11) assumes that new interaction samples are available during downstream training (referred to as the *online downstream setting*), which may incur additional costs. Given that $\mathcal{D}_{\text{pref}}$ is typically constructed from an existing offline dataset, we also consider a more practical setting: performing downstream training using only the offline dataset together with $r_{\text{new}}(s, a)$ (referred to as *the offline downstream setting*), and we adopt this setup in our experiments.[3] In this setting, an offline RL dataset is used without additional

online interaction:

$$L_{\pi_h}^{\text{offline}} = -\mathbb{E}_{\substack{(s_D, a_D) \sim \mathcal{D}_\tau \\ a \sim \pi_h \cdot \pi_l(s_D)}} \Big[Q(s_D, a) + \beta_{\text{reg}} L_{\text{reg}} + \beta_{\text{BC}} L_{\text{BC}}\Big],$$
(12)

where $\mathcal{D}_\tau$ is the offline dataset, and $L_{\text{BC}} = \|a - a_D\|_2^2$, weighted by $\beta_{\text{BC}}$, is a behavior cloning term that discourages the policy from producing out-of-distribution actions beyond the support of $\mathcal{D}_\tau$ (Fujimoto & Gu, 2021).

### 5.4. Theoretical Insights into Skill Composition

Here, we briefly outline theoretical insights into the safety and task performance of downstream policies obtained via skill composition. Formal statements and proofs are deferred to Appendix A.2. We first simplify the analysis by viewing the encoder $q(z' \mid S_z)$ as a classifier. Each trajectory is assigned to a latent variable $z'$ by the encoder and thus grouped into a subset $\mathcal{D}_{\text{pref}}^{z'}$, which is subsequently used to train a conditional utility function $\hat{u}(\cdot, z')$. Under this abstraction, we obtain two key conclusions. First, similar to Theorem 4.2, the latent-conditioned utility $\hat{u}(\cdot, z')$ learned on infinite crowd preferences prefers safe trajectories to unsafe trajectories across all $z$. As a result, if each low-level policy is optimal with respect to its corresponding conditioned utility, any downstream policy obtained via skill composition is guaranteed to preserve safety; see Corollary A.6. Moreover, we provide an upper bound on downstream safety violations when accounting for the suboptimality of low-level policies (Theorem A.7), which shows that the number of violations scales with suboptimality.

Second, we seek to explore the positive correlation between skill discovery performance and downstream task performance. Specifically, Theorems A.8 and A.9 establish a connection between (a) how well the VAE-generated labels match the true contexts and (b) how effectively the learned latent-conditioned utilities align with diverse user preferences. Part (a) characterizes the quality of skill learning, while part (b) determines the diversity of skills available for downstream task execution and thus influences final performance. In conclusion, more accurate modeling of latent variables leads to more versatile preference-aligned skills, thereby improving downstream task performance.

## 6. Safe RL Experiments

**Environments.** For evaluation, we extend six environments from the Bullet-Safety (Gronauer, 2022) and Safety-Gymnasium (Ji et al., 2023) suites to crowd preference settings by constructing a set of goal-specific reward functions, together with a common safety reward, for preference annotation and downstream tasks. In these environments, goal-specific rewards differ by task objectives (e.g., navigation targets or preferred velocities), while common safety constraints (e.g., avoiding unsafe areas or obeying speed

---

[2]Here, the high-level policy can switch skills every step. In Appendix F.3, we further discuss the effect of switching frequency.

[3]Results for the online downstream setting are also discussed in Section 6.1 and reported in Appendix D.1.

*Table 1.* Normalized performance under offline downstream settings. Overall, our methods significantly reduce safety cost compared to Task-Only baselines, while achieving task performance comparable to the Oracle.

| Environment | Stats | Baselines | | | | Ours | |
|---|---|---|---|---|---|---|---|
| | | Oracle | Task-Only | SOPL | RC($\omega = 0.5$) | Safe-VPL | Safe-CPL |
| Reach | Norm Rew | 1.00 | $1.04 \pm .00$ | $0.98 \pm .03$ | $0.83 \pm .01$ | $0.98 \pm .04$ | $0.98 \pm .04$ |
| | Norm Cost | 0.038 | $1.000 \pm .01$ | $0.024 \pm .02$ | $0.101 \pm .02$ | $0.166 \pm .04$ | $0.069 \pm .02$ |
| Run | Norm Rew | 1.00 | $1.00 \pm .00$ | $0.99 \pm .00$ | $1.00 \pm .00$ | $0.95 \pm .08$ | $0.97 \pm .01$ |
| | Norm Cost | 0.000 | $1.000 \pm .08$ | $0.000 \pm .00$ | $0.000 \pm .00$ | $0.000 \pm .00$ | $0.000 \pm .00$ |
| Circle | Norm Rew | 1.00 | $1.27 \pm .02$ | $1.04 \pm .01$ | $1.09 \pm .00$ | $0.90 \pm .01$ | $0.91 \pm .04$ |
| | Norm Cost | 0.000 | $1.000 \pm .02$ | $0.000 \pm .00$ | $0.000 \pm .00$ | $0.000 \pm .00$ | $0.000 \pm .00$ |
| Ant-vel | Norm Rew | 1.00 | $1.23 \pm .02$ | $0.98 \pm .01$ | $0.77 \pm .20$ | $0.90 \pm .01$ | $0.76 \pm .10$ |
| | Norm Cost | 0.002 | $1.000 \pm .01$ | $0.007 \pm .00$ | $0.078 \pm .12$ | $0.001 \pm .00$ | $0.007 \pm .00$ |
| Swimmer-vel | Norm Rew | 1.00 | $2.39 \pm .04$ | $1.34 \pm .03$ | $0.83 \pm .07$ | $0.88 \pm .01$ | $0.99 \pm .04$ |
| | Norm Cost | 0.000 | $1.000 \pm .09$ | $0.014 \pm .01$ | $0.000 \pm .00$ | $0.001 \pm .00$ | $0.005 \pm .00$ |
| HalfCheetah-vel | Norm Rew | 1.00 | $1.85 \pm .36$ | $0.93 \pm .00$ | $0.44 \pm .11$ | $0.96 \pm .02$ | $0.92 \pm .02$ |
| | Norm Cost | 0.000 | $1.000 \pm .19$ | $0.014 \pm .00$ | $0.107 \pm .08$ | $0.004 \pm .00$ | $0.018 \pm .02$ |
| Average | Norm Rew | 1.00 | 1.46 | 1.04 | 0.82 | 0.93 | 0.92 |
| | Norm Cost | 0.01 | 1.00 | 0.01 | 0.05 | 0.03 | 0.02 |

limits) are imposed via a shared safety reward. During preference annotation, a subset of the goal-specific rewards is selected to represent crowd variation (i.e., the user-specific component $r_{\text{user}}$) and combined with the safety reward (i.e., the shared component $r_{\text{share}}$) to form the annotation rewards. All goal-specific rewards—including those not used for annotation—are then treated as safety-agnostic task rewards, thereby defining a collection of downstream tasks.

**Dataset.** To generate preference datasets, we first train an unsafe SAC agent (Haarnoja et al., 2018) using only goal-specific reward and a safe SAC agent using the combined goal-specific and safety rewards for each designed goal. The replay buffers of these agents are combined to form the offline dataset $\mathcal{D}_\tau$. We then sample trajectory pairs to construct a crowd preference dataset, which is annotated either using Eq. 2 with the ground-truth reward or using Eq. 3 with the optimal advantage. The latter is used for CPL-based skill discovery, while the former is used for VPL-based skill discovery and other RLHF baselines. In addition, the offline dataset $\mathcal{D}_\tau$ is used for low-level policy training in Eq. 9 and offline downstream learning in Eq. 12. See more environment and dataset details in Appendix C.

**Baselines.** We evaluate three groups of baselines. **(1) Oracle.** These methods have access to the true safety reward $r_{\text{share}}$. For offline downstream learning, we apply CDT (Liu et al., 2023b), an offline safe RL algorithm, to optimize $\mathbb{E}_{\mathcal{D}_\tau}[\sum r_{\text{new}}]$ while enforcing $\mathbb{E}_{\mathcal{D}_\tau}[\sum r_{\text{share}}] = 0$, which corresponds to zero constraint violations since $r_{\text{share}} \leq 0$. For online downstream learning, we use SAC to maximize $\mathbb{E}_\pi[\sum r_{\text{new}} + r_{\text{share}}]$. **(2) Task-Only.** These baselines optimize only the task reward and ignore the safety reward using TD3+BC (Fujimoto & Gu, 2021) and TD3 (Fujimoto et al., 2018) for offline and online downstream settings, respectively. **(3) Safety-Only Preference Learning (SOPL).**

To isolate the effect of entanglement between user-specific and shared components, SOPL follows Safe RLHF (Dai et al., 2024) by using preference data generated solely from the safety reward $r_{\text{share}}$, learning a unimodal reward $\hat{r}_{\text{SOPL}}$, and optimizing $r_{\text{new}} + \hat{r}_{\text{SOPL}}$ in the downstream stage. In contrast, all other RLHF methods operate on coupled preference data without access to such separated signals. **(4) Reward Combination (RC).** RC ($\omega$) adopts TD3+BC (or TD3 for online downstream settings) to optimize a weighted combination of the downstream reward $r_{\text{new}}$ and the unimodal reward $\hat{r}$ learned from crowd preference datasets, with the trade-off weight $\omega \in \{0.1, \dots, 0.9, 1.0\}$ as defined in Eq. 5. **Safe-VPL** and **Safe-CPL** denote our proposed approaches, which use the original VPL or CPL-variant for skill discovery, respectively.

**Evaluation Protocol** We evaluate all methods in terms of both downstream task performance and shared safety performance. In the main paper, task performance is reported in normalized form: $R_{\text{norm}}(\text{Algo}) = \frac{R(\text{Algo}) - R(\text{Random})}{R(\text{Oracle}) - R(\text{Random})}$, where $R(\cdot)$ denotes the cumulative task reward $\sum_t r_{\text{new}}$ averaged over all downstream tasks. Safety performance is normalized as $C_{\text{norm}}(\text{Algo}) = \frac{C(\text{Algo})}{C(\text{Task-only baseline})}$, where $C(\cdot)$ is the cumulative safety violations. See Appendix D.2 for the main experimental results with all raw data. All experiments are run with three random seeds, and we report means and standard deviations. Code and implementation details are provided in Appendix B.

### 6.1. Recovering Shared Criteria in Crowd Preferences

Table 1 summarizes the performance of all methods under the offline downstream setting. The Task-Only baseline achieves strong task performance but incurs severe safety violations, demonstrating that optimizing an imper-

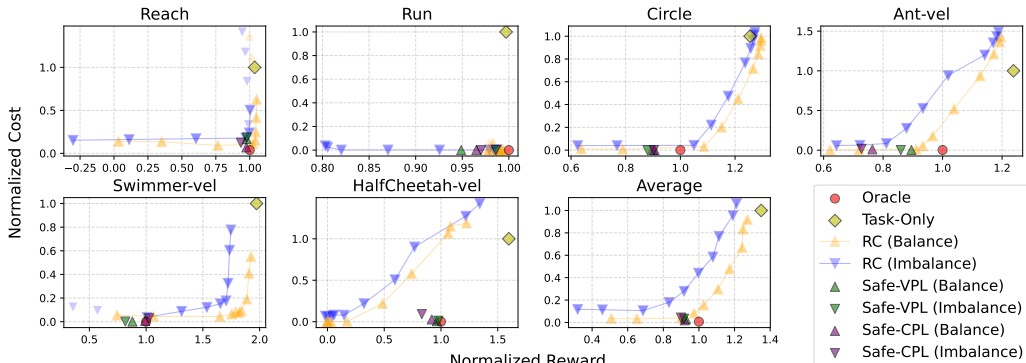

*Figure 2.* Performance points of different algorithms and RC ($\omega$) with various weight $\omega$ under balanced and imbalanced preference settings. The corresponding numerical results are reported in Table 10 in the appendix. RC is highly sensitive to the reward trade-off weight and degrades under preference imbalance, whereas our method remains consistently close to the Oracle across both settings.

fect task reward alone—without capturing underlying safety objectives—can lead to hazardous behavior. Moreover, the SOPL baseline achieves performance comparable to the Oracle, as both have access to separated safety signals, while RC($\omega = 0.5$), despite a similar training pipeline with SOPL, operates on coupled preference data and performs significantly worse, highlighting the challenge of entanglement between user-specific and shared components. In contrast, our method substantially reduces cost compared to the Task-Only baseline (e.g., 1.0 vs. 0.01–0.02 in average normalized cost), while achieving task performance comparable to both the Oracle and SOPL. This indicates that, despite not observing the true safety reward or safety-only preference data, our method can effectively discover and transfer underlying safety objectives from crowd preferences via preference-aligned skill learning and composition, without significantly sacrificing task performance.

We report results for the online downstream setting in Appendix D.1, where we observe conclusions consistent with the offline case, as well as improved sample efficiency compared to the oracle baseline trained from scratch. Moreover, performance in the online setting is generally better than in the offline case, suggesting that additional interaction further improves performance, while our offline formulation remains effective under limited interaction.

### 6.2. Comparison to Reward Combination under Balanced and Imbalanced Preference Settings

In this section, we consider both **balanced settings**, where preferences from different user-specific rewards are equally represented, and **imbalanced settings**, where one dominant user-specific preference accounts for ten times more data than the others. Figure 2 visualizes RC performance under different $\omega \in \{0.1, \dots, 1.0\}$ as reward–cost frontiers (numerical values are reported in Table 10). Across environments, RC is highly sensitive to the choice of $\omega$, often yielding either strong task performance at the expense of high safety cost, or vice versa. Moreover, under imbalanced preference data, the entire RC performance frontier shifts

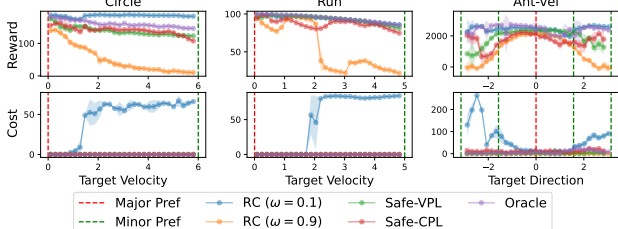

*Figure 3.* Performance on different downstream tasks under imbalanced settings; the x-axis corresponds to task IDs. RC either incurs high safety cost or performs well only on a subset of tasks, whereas our method remains robust across tasks.

noticeably away from the Oracle, consistent with the conclusion in Section 4.2 that preference imbalance induces bias in the learned reward, which in turn misguides downstream optimization and degrades task performance. In contrast, our method remains close to the Oracle under both balanced and imbalanced settings, with average normalized reward and cost decreasing by less than 0.02 across environments, whereas RC (averaged over $\omega$) degrades by at least 0.1.

While the above results summarize average performance across downstream tasks, Figure 3 highlights per-task downstream performance under imbalanced settings. When the imperfect task reward dominates ($\omega = 0.1$), RC achieves high task reward but incurs substantial safety cost on many downstream tasks; when the preference-learned reward dominates ($\omega = 0.9$), RC remains safe across all tasks but only achieves high reward on those aligned with the major preference, while degrading on minor tasks. In contrast, our method is less affected by preference imbalance, achieving strong task performance with low safety cost across tasks.

### 6.3. Ablation Studies

Figure 4 summarizes ablation studies that analyze the sensitivity of our method to key design choices. Figure 4(a) shows the effect of the prior regularization weight $\beta_{\text{reg}}$ in Eq. 11, which controls the optimization space of the high-level policy. Larger $\beta_{\text{reg}}$ enforces conservative skill selection aligned with the crowd preference distribution, yielding safer but less performant behavior, while smaller $\beta_{\text{reg}}$ in-

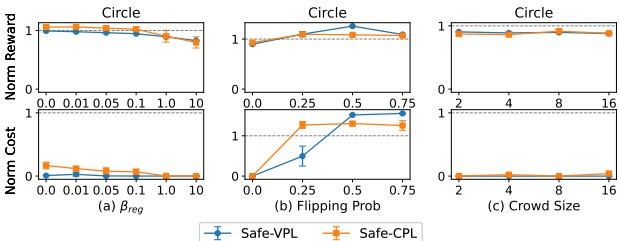

*Figure 4.* Performance under different regularization weights $\beta_{\text{reg}}$, noise ratios, and crowd sizes. Dashed lines denote $R_{\text{norm}}(\text{Oracle})$ in the first row and $C_{\text{norm}}(\text{Task-Only})$ in the second. Our method is robust to $\beta_{\text{reg}}$ and crowd size; while preference noise degrades safety performance, task reward remains largely unaffected.

|        | Task-only   | RC($\omega = 0.25$) | RC($\omega = 0.75$) | Ours          |
|--------|-------------|---------------------|---------------------|---------------|
| Reward | 0.95± .01   | 0.94 ± .01          | 0.50± .01           | 0.75 ± .01    |
| Cost   | 0.24± .00   | 0.22± .03           | 0.00 ± .00          | 0.00 ± .00    |

*Table 2.* LLM evaluation results averaged on two downstream tasks. Our method achieves strong task performance while avoiding harmful responses, whereas Task-only violates safety and RC suffers from the reward trade-off under imbalanced preferences.

creases skill flexibility and improves task performance, with only mild safety degradation that remains much better than directly optimizing the task reward. Overall, as a hyperparameter in our method, $\beta_{\text{reg}}$ is more robust than the weight $\omega$ in the RC baseline; across a wide range of values, it has little impact on task reward while consistently improving safety, making it easy to tune in practice.

An implicit assumption in our setting is that crowd preferences accurately and noiselessly reflect a shared safety consensus across all users. To analyze the impact of preference noise on our methods, we introduce preference noise by randomly flipping labels with a given probability and report results in Figure 4(b). As noise increases, the safety signal encoded in preferences is progressively corrupted, leading to degraded cost performance; however, task performance remains largely unaffected, suggesting that preference noise does not substantially reduce skill diversity leveraged for downstream composition. Figure 4(c) further analyzes the effect of the number of crowd users, where we vary crowd size by sampling target velocities or directions at regular intervals within a fixed range to construct user-specific preferences. As the number of users increases, performance exhibits only a modest degradation while maintaining substantially better safety than the task-only baseline, demonstrating the robustness of our method. See Appendix D.3 for ablation results across more environments and settings.

## 7. LLM Evaluation

While our primary evaluation focuses on continuous control, we also conduct a proof-of-concept evaluation in a simplified language-based setting to provide evidence of the broader applicability of our approach.

**Setup** Inspired by the PET dataset setup in VPL (Poddar et al., 2024), we construct a simplified RLHF-style conversational bandit setting with one query and three response categories ($A$, $B$, $C$), each containing both safe and harmful responses. Given two candidate responses as the state, the agent selects between them according to two objectives: (1) a safety objective of not selecting harmful responses, and (2) a task-specific objective of selecting the higher-ranked response when both candidates are safe. We construct an imbalanced crowd-preference dataset where User 1 (80%) and User 2 (20%) prefer safe responses over harmful ones but follow different rankings over ($A$, $B$, $C$) on safe responses.

Each downstream task is associated with a distinct ranking over response categories and provides a safety-agnostic task reward $r_{\text{new}}$ that rewards only task-specific ranking consistency, without considering the implicit safety objective. We evaluate two downstream tasks with rankings that do not match any crowd user preference ordering, meaning that simply following the majority preference or imitating a single user cannot maximize task reward. Instead, successful performance requires composing crowd preferences and generalizing beyond the preference structures observed in crowd data. See Appendix E for full experimental details.

**Results** Table 2 shows that Task-only, which optimizes only the task reward, frequently selects harmful responses to maximize reward, resulting in severe violations of the implicit safety objective. Moreover, RC either violates safety when the task reward dominates ($\omega = 0.25$) or sacrifices task performance due to bias induced by imbalanced preferences ($\omega = 0.75$). In contrast, our method substantially reduces unsafe selections compared to Task-only while achieving significantly stronger task performance than the safe but overly conservative RC baseline.

## 8. Conclusion

In this work, we focus on the hidden safety objectives embedded in crowd preferences, with the goal of discovering and transferring them to downstream tasks to encourage safe behavior. We theoretically and empirically discuss the feasibility and limitations of direct reward combination, where a reward model learned from crowd preferences is combined with the task reward for policy optimization. Then we propose a hierarchical approach that composes skills—rather than rewards—extracted from crowd preferences. Specifically, we learn latent skill representations using both a VAE reward learning and a novel CPL-based, reward-free variant, after which the high-level policy outputs latent variables for downstream tasks to optimize the task reward. We extend several safe RL environments to crowd preference settings and conduct a proof-of-concept language-domain evaluation, showing that our method achieves competitive task performance while substantially reducing safety costs, even without access to explicit safety rewards.

## Acknowledgements

This work was conducted in the Aligned, Robust, and Interactive Autonomy (ARIA) Lab at the University of Utah. ARIA Lab research is supported in part by the NSF (IIS-2310759, IIS2416761), the NIH (R21EB035378), ARPA-H, the ARL STRONG program, and Coefficient Giving.

## Impact Statement

This paper presents work whose goal is to advance the field of machine learning. One potential benefit is that our approach enables the reuse of crowd preference data across tasks, even when such preferences are not collected for a specific downstream task, potentially reducing repeated preference collection and the associated cost of human feedback. More broadly, by leveraging implicit safety signals from crowd preferences, our method may help mitigate challenges in reward design, where it is common for human-specified rewards to be incomplete or misspecified. By learning safe behaviors without explicit safety criteria, our method may facilitate RL application in domains where safety specifications are difficult to define.

At the same time, our approach assumes that crowd preference data reflect a shared consensus on safety, which may not always hold in practice due to noisy or intentionally misleading feedback. Consequently, care is required when applying this method in real-world settings, and improving robustness to misaligned or adversarial preferences remains an important direction for future work. Overall, we believe that, when applied with appropriate care, the potential benefits of our approach outweigh these limitations.

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

# A. Derivation

## A.1. Reward Combination

**Lemma A.1** (Distributional Preference Learning (Siththaranjan et al., 2024), Theorem 3.1). *When preference learning follows the Bradley–Terry formulation, the utility $\hat{u}$ learned on crowd preference implicitly aggregates hidden contexts according to the Borda count in the limit of infinite data. Formally, let $\mathcal{T}$ denote a finite trajectory set. For any two trajectories (alternatives) $a, b \in \mathcal{T}$,*

$$\hat{u}(a) > \hat{u}(b) \iff BC(a) > BC(b),$$

*where the Borda count is defined as*

$$BC(a) = \frac{1}{|\mathcal{T}|} \sum_{b \in \mathcal{T}} p_{u, D_z}(a, b), \qquad p_{u, D_z}(a, b) = \mathbb{E}_{z \sim p(z)}[O_u(a, b, z)],$$

*and*

$$O_u(a, b, z) = \begin{cases} \frac{1}{2}, & u(a, z) = u(b, z), \\ \mathbb{I}\{u(a, z) > u(b, z)\}, & \text{otherwise.} \end{cases}$$

**Lemma A.2.** *If a trajectory pair $(\tau^1, \tau^2)$ is consistent, then the learned reward model $\hat{u}$ trained on infinite crowd preference data satisfies*

$$\hat{u}(\tau^1) > \hat{u}(\tau^2) \quad or \quad \hat{u}(\tau^1) < \hat{u}(\tau^2),$$

*with the ordering matching the true utility ordering.*

*Proof.* Without loss of generality, assume $u(\tau^1, z) > u(\tau^2, z)$ for all $z$. Consider the difference between the Borda counts of $\tau^1$ and $\tau^2$:

$$BC(\tau^1) - BC(\tau^2) = \frac{1}{|\mathcal{T}|} \sum_{\tau' \in \mathcal{T}} \mathbb{E}_{z \sim p(z)}\Big[O_u(\tau^1, \tau', z) - O_u(\tau^2, \tau', z)\Big]. \tag{13}$$

We analyze the summation term-by-term.

**Case 1:** $u(\tau', z) \neq u(\tau^1, z)$ **and** $u(\tau', z) \neq u(\tau^2, z)$**.** Since $u(\tau^1, z) > u(\tau^2, z)$ for all $z$, it follows that for any $\tau'$ and any $z$,

$$O_u(\tau^1, \tau', z) = \mathbb{I}\{u(\tau^1, z) > u(\tau', z)\} \geq \mathbb{I}\{u(\tau^2, z) > u(\tau', z)\} = O_u(\tau^2, \tau', z),$$

and thus the corresponding summand is non-negative.

**Case 2:** $u(\tau', z) = u(\tau^1, z)$**.** For all $z$, we have

$$O_u(\tau^1, \tau', z) = \tfrac{1}{2}, \qquad O_u(\tau^2, \tau', z) = 0,$$

since $u(\tau^2, z) < u(\tau^1, z)$. Hence,

$$O_u(\tau^1, \tau^1, z) - O_u(\tau^2, \tau^1, z) = \tfrac{1}{2} > 0.$$

**Case 3:** $u(\tau', z) = u(\tau^2, z)$**.** For all $z$, we have

$$O_u(\tau^1, \tau', z) = 1, \qquad O_u(\tau^2, \tau', z) = \tfrac{1}{2},$$

and therefore

$$O_u(\tau^1, \tau^2, z) - O_u(\tau^2, \tau^2, z) = \tfrac{1}{2} > 0.$$

Combining the above cases, every term in the summation is non-negative, and at least two terms are strictly positive ($\tau^1$ and $\tau^2$ themselves). Therefore,

$$BC(\tau^1) - BC(\tau^2) > 0.$$

According to Lemma A.1, the learned reward model $\hat{u}$ recovers the ordering induced by the Borda counts (Lemma A.1), and therefore

$$\hat{u}(\tau^1) > \hat{u}(\tau^2).$$

$\square$

**Theorem A.3.** *Let the true utility be $u(\tau, z) = r_{\text{user}}(s, a, z) + r_{\text{share}}(s, a)$, where the shared component $r_{\text{share}}(s, a)$ represents a safety penalty:*

$$r_{\text{share}}(s, a) = \begin{cases} -K, & (s, a) \in X_{\text{unsafe}}, \\ 0, & \text{otherwise.} \end{cases}$$

*If the penalty $K > 2L \max_{s,a,z} |r_{\text{user}}(s, a, z)|$, then all pairs $(\tau^{\text{safe}}, \tau^{\text{unsafe}})$ consisting of one safe and one unsafe trajectory are consistent. Consequently, the single reward model $\hat{u}$ learned from infinite crowd preference data satisfies*

$$\hat{u}(\tau^{\text{safe}}) > \hat{u}(\tau^{\text{unsafe}}),$$

*for any $\tau^{\text{safe}}$ and any $\tau^{\text{unsafe}}$.*

*Proof.* For any pair $(\tau^{\text{safe}}, \tau^{\text{unsafe}})$, by definition there exists at least one unsafe state-action pair $(s, a) \in X_{\text{unsafe}}$ in $\tau^{\text{unsafe}}$. We consider the utility difference

$$u(\tau^{\text{safe}}, z) - u(\tau^{\text{unsafe}}, z) \tag{14}$$

$$= \sum_t \left[ r_{\text{user}}(s_t^{\text{safe}}, a_t^{\text{safe}}, z) - r_{\text{user}}(s_t^{\text{unsafe}}, a_t^{\text{unsafe}}, z) \right] + \sum_t [0 - r_{\text{share}}(s_t^{\text{unsafe}}, a_t^{\text{unsafe}})] \tag{15}$$

$$\geq - \left( \sum_t 2 \max_{s,a,z} |r_{\text{user}}(s, a, z)| \right) + K. \tag{16}$$

$$\tag{17}$$

Therefore, if $K > 2L \max_{s,a,z} |r_{\text{user}}(s, a, z)|$, we have

$$u(\tau^{\text{safe}}, z) > u(\tau^{\text{unsafe}}, z) \quad \text{for all } z,$$

which implies that every safe–unsafe trajectory pair is consistent. According to Lemma A.2, we have $\hat{u}(\tau^{\text{safe}}) > \hat{u}(\tau^{\text{unsafe}})$ for any safe–unsafe pair.

$\square$

**Theorem A.4.** *Assume a finite set of hidden contexts $\{z_1, \ldots, z_m\}$ and a finite trajectory set $\mathcal{T}$. Denote $X_{\text{ics}}$ as the set of all inconsistent pairs, define*

$$N(\tau^1, \tau^2, z) = \sum_{\tau \in \mathcal{T}} \mathbb{I}\{u(\tau^1, z) > u(\tau, z) > u(\tau^2, z)\} +$$
$$\mathbb{I}\{u(\tau^1, z) < u(\tau, z) < u(\tau^2, z)\}.$$

*For any $z_k$, if the probability of $z_k$ in the crowd preference dataset satisfies*

$$p(z_k) > \frac{|\mathcal{T}| - 1}{\min_{(\tau, \tau') \in X_{\text{ics}}} N(\tau, \tau', z_k) + |\mathcal{T}|}$$

*then for all $\tau^1, \tau^2 \in \mathcal{T}$,*

$$\hat{u}(\tau^1) > \hat{u}(\tau^2) \iff u(\tau^1, z_k) > u(\tau^2, z_k).$$

*Proof.* From the definition of the Borda count, for all $\tau^1, \tau^2 \in \mathcal{T}$,

$$BC(\tau^1) - BC(\tau^2) \tag{18}$$

$$= \frac{1}{|\mathcal{T}|} \sum_i p(z_i) \sum_{\tau \in \mathcal{T}} \left[ O_u(\tau^1, \tau, z_i) - O_u(\tau^2, \tau, z_i) \right]. \tag{19}$$

Denote

$$S_i := \sum_{\tau \in \mathcal{T}} \left[ O_u(\tau^1, \tau, z_i) - O_u(\tau^2, \tau, z_i) \right],$$

so that $BC(\tau^1) - BC(\tau^2) = \frac{1}{|\mathcal{T}|} \sum_i p(z_i) S_i$. We now analyze the bounds of $S_i$. For an arbitrary context $z_i$, assume without loss of generality that $u(\tau^1, z_i) > u(\tau^2, z_i)$. Consider each trajectory $\tau \in \mathcal{T}$ in five cases:

- **Case 1:** $u(\tau, z_i) > u(\tau^1, z_i)$. Then
$$O_u(\tau^1, \tau, z_i) = O_u(\tau^2, \tau, z_i) = 0,$$
so the contribution is 0.

- **Case 2:** $u(\tau, z_i) = u(\tau^1, z_i)$. Then
$$O_u(\tau^1, \tau, z_i) = \tfrac{1}{2}, \qquad O_u(\tau^2, \tau, z_i) = 0,$$
so the contribution is $\tfrac{1}{2}$. In particular, the self-comparison term $\tau = \tau^1$ contributes $\tfrac{1}{2}$.

- **Case 3:** $u(\tau^1, z_i) > u(\tau, z_i) > u(\tau^2, z_i)$. Then
$$O_u(\tau^1, \tau, z_i) = 1, \qquad O_u(\tau^2, \tau, z_i) = 0,$$
so the contribution is 1. There are exactly $N(\tau^1, \tau^2, z_i)$ such trajectories.

- **Case 4:** $u(\tau, z_i) = u(\tau^2, z_i)$. Then
$$O_u(\tau^1, \tau, z_i) = 1, \qquad O_u(\tau^2, \tau, z_i) = \tfrac{1}{2},$$
so the contribution is $\tfrac{1}{2}$. In particular, the self-comparison term $\tau = \tau^2$ contributes $\tfrac{1}{2}$.

- **Case 5:** $u(\tau, z_i) < u(\tau^2, z_i)$. Then
$$O_u(\tau^1, \tau, z_i) = O_u(\tau^2, \tau, z_i) = 1,$$
so the contribution is 0.

Therefore, Case 3 occurs for exactly $N(\tau^1, \tau^2, z_i)$ trajectories, while Cases 2 and 4 together contribute at least 1 due to the self-comparisons $\tau = \tau^1$ and $\tau = \tau^2$. Hence,
$$S_i = \sum_{\tau \in \mathcal{T}} \left[ O_u(\tau^1, \tau, z_i) - O_u(\tau^2, \tau, z_i) \right] \geq 1 + N(\tau^1, \tau^2, z_i).$$

Moreover, since every term is at most 1 and the remaining $|\mathcal{T}| - 2$ non-self comparisons can contribute at most 1 each, we have
$$S_i = \sum_{\tau \in \mathcal{T}} \left[ O_u(\tau^1, \tau, z_i) - O_u(\tau^2, \tau, z_i) \right] \leq |\mathcal{T}| - 1.$$

Hence,
$$1 + N(\tau^1, \tau^2, z_i) \leq S_i \leq |\mathcal{T}| - 1.$$

By symmetry, when
$$u(\tau^1, z_i) < u(\tau^2, z_i),$$
the ordering of all comparisons is reversed, yielding
$$-(|\mathcal{T}| - 1) \leq S_i \leq -\left(1 + N(\tau^1, \tau^2, z_i)\right).$$

Moreover, if $u(\tau^1, z_i) = u(\tau^2, z_i)$, then for every $\tau \in \mathcal{T}$ we have $O_u(\tau^1, \tau, z_i) = O_u(\tau^2, \tau, z_i)$, and hence $S_i = 0$.

Now we aim to derive the worst-case lower bound of $BC(\tau^1) - BC(\tau^2)$. Without loss of generality, assume
$$u(\tau^1, z_k) > u(\tau^2, z_k),$$
so that
$$S_k \geq 1 + N(\tau^1, \tau^2, z_k).$$

For any other context $z_i \neq z_k$, the relative ordering between $u(\tau^1, z_i)$ and $u(\tau^2, z_i)$ can be arbitrary. Therefore, in the worst case,
$$S_i \geq -(|\mathcal{T}| - 1), \qquad i \neq k.$$

Combining these bounds, we obtain

$$BC(\tau^1) - BC(\tau^2) = \frac{1}{|\mathcal{T}|} \sum_i p(z_i) S_i \geq \frac{1}{|\mathcal{T}|} \left[ p(z_k)\big(1 + N(\tau^1, \tau^2, z_k)\big) - (1 - p(z_k))(|\mathcal{T}| - 1) \right].$$

Hence, if

$$p(z_k) > \frac{|\mathcal{T}| - 1}{N(\tau^1, \tau^2, z_k) + |\mathcal{T}|},$$

then $BC(\tau^1) - BC(\tau^2) > 0$, which implies

$$\hat{u}(\tau^1) > \hat{u}(\tau^2).$$

Therefore, when $p(z_k) > \frac{|\mathcal{T}|-1}{N(\tau^1,\tau^2,z_k)+|\mathcal{T}|}$, we have

$$u(\tau^1, z_k) > u(\tau^2, z_k) \iff BC(\tau^1) - BC(\tau^2) > 0 \iff \hat{u}(\tau^1) > \hat{u}(\tau^2)$$

Let

$$p(z_k) > \max_{(\tau,\tau') \in X_{\mathrm{ics}}} \frac{|\mathcal{T}| - 1}{N(\tau, \tau', z_k) + |\mathcal{T}|} = \frac{|\mathcal{T}| - 1}{\min_{(\tau,\tau') \in X_{\mathrm{ics}}} N(\tau, \tau', z_k) + |\mathcal{T}|},$$

then for every inconsistent pair $(\tau^1, \tau^2) \in X_{\mathrm{ics}}$ satisfying

$$u(\tau^1, z_k) > u(\tau^2, z_k),$$

we have $p(z_k) > \max_{(\tau,\tau') \in X_{\mathrm{ics}}} \frac{|\mathcal{T}|-1}{N(\tau,\tau',z_k)+|\mathcal{T}|} > \frac{|\mathcal{T}|-1}{N(\tau^1,\tau^2,z_k)+|\mathcal{T}|}$ and thus

$$\hat{u}(\tau^1) > \hat{u}(\tau^2).$$

Since consistent pairs share the same ordering across all contexts, while every inconsistent pair follows the ordering induced by $z_k$, the learned reward ordering induced by $\hat{u}$ coincides exactly with the ordering induced by $u(\cdot, z_k)$ over all trajectory pairs.

Moreover, if we assume every trajectory has a distinct utility, we have $S_i = 1 + N(\tau^1, \tau^2, z_i)$ or $S_i = -(1 + N(\tau^1, \tau^2, z_i))$, and thus

$$BC(\tau^1) - BC(\tau^2) \geq \frac{1}{|\mathcal{T}|} \left[ p(z_k)\big(1 + N(\tau^1, \tau^2, z_k)\big) - (1 - p(z_k))\big(1 + \max_i N(\tau^1, \tau^2, z_i)\big) \right].$$

Finally, we can yield a tighter bound

$$p(z_k) > \max_{(\tau,\tau') \in X_{\mathrm{ics}}} \frac{1 + \max_i N(\tau, \tau', z_i)}{2 + N(\tau, \tau', z_k) + \max_i N(\tau, \tau', z_i)}$$

$$\square$$

## A.2. Policy Composition

**Corollary A.5.** *For any context $z_k$, the learned utility function prefers safe trajectories over unsafe ones, i.e.,*

$$\hat{u}(\tau_{\mathrm{safe}}, z_k') > \hat{u}(\tau_{\mathrm{unsafe}}, z_k'),$$

*for any $z_k'$ and any pair of trajectories $\tau_{\mathrm{safe}}$ and $\tau_{\mathrm{unsafe}}$ in the limit of infinite data.*

This result follows from the fact that each conditioned utility $\hat{u}(\cdot, z_k')$ is trained only on a subset of the crowd preference dataset $\mathcal{D}_{\mathrm{pref}}$, consisting of trajectories assigned to the latent variable $z_k'$ by the encoder $q$, i.e., $\mathcal{D}^{z_k'} = \{\tau \mid q(z_k' \mid S_z) \neq 0, \tau \in S_z, S_z \in \mathcal{D}_{\mathrm{pref}}\}$. As established in the Theorem 4.2, all safe-unsafe trajectory pairs in $\mathcal{D}_{\mathrm{pref}}$ are consistent. Therefore, all safe-unsafe pairs in $D^{z_k'}$, which is a subset of $\mathcal{D}_{\mathrm{pref}}$, should remain consistent, and thus the learned conditioned utility $\hat{u}(\cdot, z_k')$ maintains the ordering between safe and unsafe trajectories, yielding $\hat{u}(\tau_{\mathrm{safe}}, z_k') > \hat{u}(\tau_{\mathrm{unsafe}}, z_k')$.

Next, we show that composing low-level policies—each of which is optimal with respect to its conditioned utility—guarantees safety.

**Corollary A.6.** *If each low-level policy $\pi(\cdot \mid s, z_k)$ is optimal with respect to the conditioned utility $\hat{u}(\cdot, z_k)$, then any downstream policy obtained by composing the low-level policies $\{\pi(\cdot \mid s, z_k)\}$ satisfies the safety criteria.*

*Proof.* By construction, the downstream policy operates by selecting a latent variable $z_k$ and executing the corresponding low-level policy $\pi(\cdot \mid s, z_k)$ for a finite horizon. Therefore, any trajectory $\tau$ induced by the downstream policy can be expressed as a concatenation of trajectory segments

$$\tau = \tau^{(1)} \circ \tau^{(2)} \circ \cdots \circ \tau^{(m)},$$

where each segment $\tau^{(i)}$ is generated by some low-level policy $\pi(\cdot \mid s, z_{k_i})$.

From Corollary A.5, each conditioned utility $\hat{u}(\cdot, z_{k_i})$ strictly prefers safe trajectories over unsafe ones. Consequently, the corresponding low-level policy $\pi(\cdot \mid s, z_{k_i})$, which is optimal with respect to $\hat{u}(\cdot, z_{k_i})$, induces only safe trajectories if safe trajectories are feasible; that is, all state–action pairs along each segment $\tau^{(i)}$ satisfy the safety constraints.

Since safety violations are defined at the level of state–action pairs, and each segment $\tau^{(i)}$ is safe, their concatenation $\tau$ cannot contain any unsafe state–action pair. Hence, any downstream policy obtained by composing the low-level policies is guaranteed to be safe. $\qquad\square$

**Theorem A.7.** *Consider a set of low-level policies $\{\pi(\cdot \mid s, z)\}$, where each policy $\pi(\cdot \mid s, z)$ is $\delta_z$-suboptimal with respect to its conditioned utility $\hat{u}(\cdot, z)$, i.e.,*

$$\mathbb{E}_{\pi^*(\cdot \mid z)}[\hat{u}] - \mathbb{E}_{\pi(\cdot \mid z)}[\hat{u}] \leq \delta_z,$$

*where $\pi^*(\cdot \mid z)$ denotes the optimal policy under $\hat{u}(\cdot, z)$.*

*Assume the safety penalty is $K > 0$ per violation, and the downstream policy composes at most $m$ low-level policies per trajectory. Then the expected number of safety violations incurred by the downstream policy is bounded by*

$$N(violation) \leq \frac{1}{K} \sum_{j=1}^{m} \delta_{z_j} \leq \frac{m \max_z \delta_z}{K}.$$

*Proof.* Consider a low-level policy $\pi(\cdot \mid z)$ with utility gap $\delta_z$ relative to the optimal policy $\pi^*(\cdot \mid z)$. The utility is defined as $\sum_t r_{\text{user}} + r_{\text{share}}$, where violations of safety constraints incur a penalty of $K$ per occurrence.

In the worst case, the entire utility gap $\delta_z$ can be attributed to safety violations. Since each violation contributes at least $K$ to the utility difference, the expected number of violations under $\pi(\cdot \mid z)$ is bounded by $\delta_z / K$.

Now consider a downstream policy that composes at most $m$ low-level policies, yielding at most $m$ trajectory segments, each governed by some $\pi(\cdot \mid z_j)$. Summing over all segments, the total number of violations is bounded by

$$N(\text{violation}) \leq \sum_{j=1}^{m} \frac{\delta_{z_j}}{K} = \frac{1}{K} \sum_{j=1}^{m} \delta_{z_j}.$$

Finally, since $\delta_{z_j} \leq \max_z \delta_z$ for all $j$, we obtain

$$N(\text{violation}) \leq \frac{m \max_z \delta_z}{K}.$$

$\qquad\square$

We next study the downstream task performance achieved by policies composed from learned skills. Although a direct analysis is challenging, it is clear that downstream performance is strongly influenced by the diversity of the learned skills. In particular, if the VAE collapses most preferences into a single latent subset, the model effectively learns only one skill, which typically lacks the behavioral primitives required to solve new downstream tasks. Therefore, desirable skills should align with a diverse set of underlying preferences in the crowd preference dataset, thereby exhibiting sufficient diversity.

To facilitate analysis, we introduce two simplifying assumptions. First, we assume that the encoder $q_\psi(z' \mid S_z)$ outputs a categorical distribution over $m$ annotation tasks, predicting which task each preference set $S_z$ originates from.[4] We further

---

[4]In practice, $m$ is unknown, and we therefore use a continuous encoder with Gaussian outputs rather than a categorical encoder.

assume that the probability of misassignment is upper bounded by $\epsilon$. Under this assumption, Theorem A.8 shows that for any two trajectories, if their utility-based ranking gap in context $z_k$ exceeds a threshold depending on $\epsilon$, then their preference relationship induced by the learned utility is consistent with the true utility. In the special case $\epsilon = 0$, the learned utility recovers the correct ordering over all trajectories, implying that the resulting low-level policy $\pi_l(a \mid s, z'_k)$ will be optimal for context $z_k$. Second, although the learned utility derived from the Bradley–Terry model does not have a closed-form expression, we leverage the equivalence between Borda count and the learned utilities in terms of ranking, and thus approximate the learned utility using Borda count, which in turn enables a tractable analysis of the KL divergence between the trajectory distribution induced by $\pi_l(a \mid s, z'_k)$ and that of the optimal policy for context $z_k$. As shown in Theorem A.9, this divergence is upper bounded by $\epsilon^2$.

Together, Theorems A.8 and A.9 establish a connection between the quality of the skill discovery—captured by the classification error $\epsilon$ of VAE encoder—and the optimality of the learned utilities and low-level policies under the corresponding context. Since the latter directly influence skill diversity, these results provide theoretical insight into how skill discovery influences downstream task performance through skill composition.

**Theorem A.8.** *Assume that under the hidden context $z_k$, all trajectories are ranked by their true utilities $u(\tau, z_k)$ such that*

$$u(\tau^1, z_k) \leq u(\tau^2, z_k) \leq \cdots \leq u(\tau^{|\mathcal{T}|}, z_k).$$

*Let $\epsilon \in [0,1]$ be an upper bound on the probability that a sample assigned to $z_k$ actually has a different true context, i.e., $P(\text{True} \neq z_k | \text{Assigned} = z_k) \leq \epsilon$. If for two trajectories $\tau^i$ and $\tau^j$, the rank gap satisfies*

$$i - j > |\mathcal{T}| \frac{\epsilon}{1 - \epsilon},$$

*then under the learned model we have*

$$\hat{u}(\tau^i, z_k) > \hat{u}(\tau^j, z_k).$$

*Proof.* Under our assumption that $P(\text{True} \neq z_k | \text{Assigned} = z_k) \leq \epsilon$, the true context distribution of preferences within this training set $D^{z'_k}$ can be written as

$$P'_{z_k}(z) = (1 - \epsilon)\mathbb{I}\{z = z_k\} + \epsilon R(z),$$

where $R(z)$ denotes an arbitrary distribution over contexts corresponding to misassigned preference data.

For any trajectory $\tau$, the Borda counts under the $D^{z'_k}$ are given respectively by

$$BC'_{z_k}(\tau) = \frac{1}{|\mathcal{T}|} \sum_{\tau'} \mathbb{E}_{z \sim P'_{z_k}(z)}[O_u(\tau, \tau', z)], \tag{20}$$

$$= (1 - \epsilon)BC_{z_k}(\tau) + \epsilon \tilde{BC}_{z_k}(\tau), \tag{21}$$

where

$$BC_{z_k}(\tau) = \frac{1}{|\mathcal{T}|} \sum_{\tau'} O_u(\tau, \tau', z_k), \qquad \tilde{BC}_{z_k}(\tau) = \frac{1}{|\mathcal{T}|} \sum_{\tau'} \mathbb{E}_{z \sim R(z)}[O_u(\tau, \tau', z)]$$

Now consider two trajectories $\tau^i$ and $\tau^j$ with ranks $i > j$ in the true order, i.e. $BC_{z_k}(\tau^i) > BC_{z_k}(\tau^j)$. For convenience, let

$$\Delta_{ij} = BC_{z_k}(\tau^i) - BC_{z_k}(\tau^j) = \frac{1}{|\mathcal{T}|}\left[\sum_{\tau} O_u(\tau^i, \tau, z_k) - O_u(\tau^j, \tau, z_k)\right].$$

Since $\tau^i$ is ranked at position $i$ and $\tau^j$ is ranked at position $j$, there are at least $i - j + 1$ trajectories whose utilities lie between them. Besides, at least one trajectories ($\tau^i$ / $\tau^j$) itself) share the same utility with $\tau^i$ / $\tau^j$. Therefore,

$$\sum_{\tau} \left[O_u(\tau^i, \tau, z_k) - O_u(\tau^j, \tau, z_k)\right]$$

$$\leq [O_u(\tau^i, \tau^i, z_k) + 0] + \sum_{j < l < i} \left[O_u(\tau^i, \tau^l, z_k) - O_u(\tau^j, \tau^l, z_k)\right] + [O_u(\tau^i, \tau^j, z_k) - O_u(\tau^j, \tau^j, z_k)] = \frac{i - j}{|\mathcal{T}|}.$$

Then, since $O_u(\cdot) \in [0,1]$, we have $0 \leq \tilde{BC}_{z_k}(\tau) \leq 1$ and $0 \leq BC_{z_k}(\tau) \leq 1$ for all $\tau$. We can bound the noisy Borda counts as follows:

$$BC'_{z_k}(\tau^i) = (1-\epsilon)BC_{z_k}(\tau^i) + \epsilon \tilde{BC}_{z_k}(\tau^i) \geq (1-\epsilon)BC_{z_k}(\tau^i),$$

$$BC'_{z_k}(\tau^j) = (1-\epsilon)BC_{z_k}(\tau^j) + \epsilon \tilde{BC}_{z_k}(\tau^j) \leq (1-\epsilon)BC_{z_k}(\tau^j) + \epsilon.$$

Hence,

$$BC'_{z_k}(\tau^i) - BC'_{z_k}(\tau^j) \geq (1-\epsilon)\big(BC_{z_k}(\tau^i) - BC_{z_k}(\tau^j)\big) - \epsilon \geq (1-\epsilon)\frac{i-j}{|\mathcal{T}|} - \epsilon.$$

If $i - j > |\mathcal{T}|\frac{\epsilon}{1-\epsilon}$ satisfies, then

$$BC'_{z_k}(\tau^i) - BC'_{z_k}(\tau^j) > 0$$

Under this condition, the perturbed Borda order is consistent with the true order, implying that the learned model preserves the relative preference between $\tau^i$ and $\tau^j$ according to Lemma A.1:

$$BC'_{z_k}(\tau^i) > BC'_{z_k}(\tau^j) \implies \hat{u}(\tau^i, z_k) > \hat{u}(\tau^j, z_k).$$

$\square$

**Theorem A.9.** *Assume that under the hidden context $z_k$, the learned utility is approximated by Borda count. Let $\epsilon \in [0,1]$ be an upper bound on the probability that a sample assigned to $z_k$ actually has a different true context satisfies Under the MaxEnt framework with the entropy temperature $\alpha'$, let $p_{z_k}(\tau) \propto \exp\left(\frac{1}{\alpha'}BC(\tau)\right)$ denote the approximated optimal trajectory distribution for the utility $u(\cdot, z_k)$ under context $z_k$, and let $q_{z_k}(\tau) \propto \exp\left(\frac{1}{\alpha'}BC'(\tau)\right)$ denote the approximate distribution for the learned utility $\hat{u}(\cdot, z_k)$, where $BC(\tau)$ and $BC'(\tau)$ is the Borda count of $\tau$ under the $u(\cdot, z_k)$ and $\hat{u}(\cdot, z_k)$, respectively. Then the Kullback–Leibler divergence admits the bound*

$$\mathrm{KL}\big(p_{z_k} \,\|\, q_{z_k}\big) \leq \frac{\epsilon^2}{2\alpha'^2}.$$

*Proof.* From the proof of Theorem A.8, for any trajectory $\tau$, the Borda counts under the $D^{z'_k}$ are given respectively by

$$BC'_{z_k}(\tau) = \frac{1}{|\mathcal{T}|} \sum_{\tau'} \mathbb{E}_{z \sim P'_{z_k}(z)}[O_u(\tau, \tau', z)], \tag{22}$$

$$= (1-\epsilon)BC_{z_k}(\tau) + \epsilon \tilde{BC}_{z_k}(\tau), \tag{23}$$

where

$$BC_{z_k}(\tau) = \frac{1}{|\mathcal{T}|} \sum_{\tau'} O_u(\tau, \tau', z_k), \qquad \tilde{BC}_{z_k}(\tau) = \frac{1}{|\mathcal{T}|} \sum_{\tau'} \mathbb{E}_{z \sim R(z)}[O_u(\tau, \tau', z)]$$

Let

$$\Delta(\tau) := BC'_{z_k}(\tau) - BC_{z_k}(\tau) = \epsilon\big(\tilde{BC}_{z_k}(\tau) - BC_{z_k}(\tau)\big).$$

Since each $O_u(\cdot) \in [0,1]$, both $BC_{z_k}(\tau)$ and $\tilde{BC}_{z_k}(\tau)$ lie in $[0,1]$, and hence

$$|\Delta(\tau)| \leq \epsilon \quad \text{for all } \tau.$$

Under the maximum entropy RL framework, the optimal trajectory distribution induced by a reward function $r$ is given by $p(\tau) = \exp\left(\frac{1}{\alpha'}u(\tau)\right)$ (Levine, 2018), where $u(\tau)$ denotes the utility (or cumulative reward) and $\alpha'$ is temperature parameter. Since the trajectory-level utility learned via the Bradley-Terry loss does not admit a closed-form expression, we exploit the ranking equivalence between the learned utility and the Borda count to approximate the utility, i.e., $u(\tau, z) \approx BC_z(\tau)$. Accordingly, we approximate the optimal trajectory distributions induced by the true utility $u(\cdot, z_k)$ and the learned utility $\hat{u}(\cdot, z_k)$ as follows:

$$p(\tau) = \frac{\exp \frac{1}{\alpha'}BC_{z_k}(\tau)}{Z}, \qquad q(\tau) = \frac{\exp \frac{1}{\alpha'}BC'_{z_k}(\tau)}{Z'},$$

with partition functions $Z = \sum_\tau \exp \frac{1}{\alpha'}BC_{z_k}(\tau)$ and $Z' = \sum_\tau \exp \frac{1}{\alpha'}BC'_{z_k}(\tau)$. It is worth noting that when the temperature parameter $\alpha'$ approaches 0, the above approximation becomes exactly equivalent to the true optimal trajectory,

since probability mass is assigned exclusively to trajectories that achieve the maximum utility, which coincide with those achieving the highest Borda count.

Compute the KL divergence:

$$\mathrm{KL}(p\|q) = \sum_\tau p(\tau) \log \frac{p(\tau)}{q(\tau)} = \sum_\tau p(\tau)\Big(\frac{1}{\alpha'}BC_{z_k}(\tau) - \frac{1}{\alpha'}BC'_{z_k}(\tau)\Big) + \log \frac{Z'}{Z}$$

$$= -\frac{1}{\alpha'}\mathbb{E}_p[\Delta] + \log \frac{Z'}{Z}.$$

Noting that

$$Z' = \sum_\tau \exp\frac{BC'_{z_k}(\tau)}{\alpha'} = \sum_\tau \exp\frac{BC_{z_k}(\tau)}{\alpha'}\exp\frac{\Delta(\tau)}{\alpha'} = Z\sum_\tau \frac{\exp\frac{BC_{z_k}(\tau)}{\alpha'}}{Z}\exp\frac{\Delta(\tau)}{\alpha'} = Z\,\mathbb{E}_p\big[\exp\frac{\Delta(\tau)}{\alpha'}\big],$$

we have

$$\log\frac{Z'}{Z} = \log\mathbb{E}_p\big[\exp\frac{\Delta(\tau)}{\alpha'}\big].$$

Therefore the KL divergence can be written exactly as

$$\mathrm{KL}(p\|q) = -\mathbb{E}_p[\frac{\Delta(\tau)}{\alpha'}] + \log\mathbb{E}_p\big[\exp\frac{\Delta(\tau)}{\alpha'}\big].$$

Now after applying Hoeffding's lemma to the bounded random variable satisfying $\Delta(\tau)/\alpha' \in [-\epsilon/\alpha', \epsilon/\alpha']$ under distribution $p$, we have

$$\log\mathbb{E}_p\big[\exp\frac{\Delta(\tau)}{\alpha'}\big] \le \mathbb{E}_p[\frac{\Delta(\tau)}{\alpha'}] + \frac{(2\epsilon)^2}{8\alpha'^2} = \mathbb{E}_p[\frac{\Delta(\tau)}{\alpha'}] + \frac{\epsilon^2}{2\alpha'^2}.$$

Substituting this bound into the expression for the KL divergence yields

$$\mathrm{KL}(p\|q) \le -\mathbb{E}_p[\frac{\Delta(\tau)}{\alpha'}] + \Big(\mathbb{E}_p[\frac{\Delta(\tau)}{\alpha'}] + \frac{\epsilon^2}{2\alpha'^2}\Big) = \frac{\epsilon^2}{2\alpha'^2}.$$

which completes the proof. □

## B. Implementation Details

### B.1. Our Method

Our code is available at https://github.com/qianlin04/Implicit-Safety-Alignment-from-Crowd-Preferences.

**Skill Discovery** Our implementation is based on the official VPL codebase[5]. We consider two variants in this work. The first variant follows the original VPL formulation: we first learn a latent-conditioned reward function $r_\phi(s, a, z')$ using the VPL losses in Eqs. 7,8, and then train a latent-conditioned low-level policy $\pi_\theta(a \mid s, z')$ via IQL (Kostrikov et al., 2022). The second variant combines VAE-based loss in Eq. 7 with CPL (i.e., Eq. 10), to directly learn the latent-conditioned policy $\pi_\theta(a \mid s, z')$. However, we find that this variant jointly training the encoder $q_\psi(z' \mid S_z)$ and the policy $\pi_\theta(a \mid s, z')$ leads to unstable policy learning. To address this issue, we first jointly train the encoder and the latent-conditioned reward using the standard VPL losses in Eqs. 7 and 8, following the first variant. We then discard the latent-conditioned reward, fix the encoder and update only the policy using the objectives in Eqs. 7 and 10. Unless otherwise specified, we adopt the default hyperparameters from the VPL and CPL implementations. Hyperparameters that differ from the original implementations are summarized in Table 3.

---
[5]https://github.com/WEIRDLabUW/vpl

**Downstream Training** We train the downstream policy using TD3 and TD3+BC for the online and offline downstream settings, respectively. Our implementations are based on the official codebases[6]. To further prevent the exploitation of out-of-distribution skills, in addition to the prior regularization term, we explicitly constrain the output range of the high-level policy. Specifically, we apply a $\texttt{tanh}$ activation at the output of the policy network, yielding the high-level action

$$a_h = A_{\max} \cdot \tanh(x),$$

where $x$ denotes the output of the final layer of the policy network, and $A_{\max}$ is a manually specified bound. The resulting latent variable is constructed as

$$z' = \mathrm{Mean}(p(z')) + \mathrm{Std}(p(z')) \cdot a_h,$$

which is then used as the input to the low-level policy $\pi_\theta(a \mid s, z')$ to produce the final environment action. The specific hyperparameters used for downstream training are summarized in Table 3.

### B.2. Baselines

TD3 and TD3+BC also serve as the backbone algorithms for both the Task-only and RC baselines. These baselines directly learn action-level policies that output environment actions, whereas our approach optimizes a high-level policy while keeping the low-level policy fixed. The Task-only baseline optimizes only the downstream task reward, consistent with our objective, while the RC baseline optimizes a fused reward that combines the downstream task reward with a single reward learned from the crowd preference dataset. In addition, we adopt CDT as the oracle baseline under the offline downstream setting, implemented based on the OSRL framework[7].

**Constraint Decision Transformer (CDT)** CDT has access to real safety reward, and solves the following constrained optimization problem:

$$\mathbb{E}_{\mathcal{D}_\tau}[r_{\mathrm{new}}(s_t, a_t)] \text{ s.t. } \mathbb{E}_{\mathcal{D}_\tau}[r_{\mathrm{share}}(s_t, a_t)] = 0.$$

Specifically, CDT employs a transformer-based policy trained via trajectory cloning:

$$\max_\pi \; \mathbb{E}_{\tau \sim \mathcal{D}_\tau} \left[ \sum_{t=1}^{T} \log \pi(a_t \mid s_t, G_t^r, G_t^c) \right],$$

where $G_t^r$ denotes the return-to-go from step $t$ to $T$ (i.e., $G_t^r = \sum_{i=t}^{T} r_{\mathrm{new},i}$), and $G_t^c$ is a randomly sampled constraint threshold that upper-bounds the cumulative cost from $t$ to $T$ (i.e., $G_t^c \geq \sum_{i=t}^{T} r_{\mathrm{share},i}$). During evaluation, we set $G_t^c = 0$ to enforce constraint satisfaction, and set $G_t^r$ to the task performance achieved by the oracle (SAC trained with full reward).

**Reward Combine (RC)** For the RC baselines, we first learn a single reward model $r_\phi(s, a)$ from the crowd preference dataset using Eq. 2, and then combine it with the new-task reward $r_{\mathrm{new}}(s, a)$ to form the downstream objective. We observe that the scale of $r_\phi(s, a)$ is often significantly larger than that of $r_{\mathrm{new}}(s, a)$. To mitigate this issue, we adopt two techniques: (1) we add an $L_2$ regularization term on the reward outputs in the Bradley–Terry loss when training $r_\phi$:

$$L_\phi = \mathbb{E}_{(\tau^1, \tau^2) \sim \mathcal{D}_{\mathrm{pref}}} \left[ y \cdot \log P(y = 1 \mid \tau^1, \tau^2) + (1 - y) \cdot \log P(y = 0 \mid \tau^1, \tau^2) + \kappa \cdot L_2(r_\phi) \right],$$

where $L_2(r_\phi)$ denotes the $\ell_2$ norm of the predicted rewards over $\tau^1$ and $\tau^2$; (2) we normalize $r_\phi$ over the entire offline dataset $\mathcal{D}_\tau$ by subtracting the mean and dividing by the standard deviation.

## C. Safe RL Experiment Details

### C.1. Environment

The Reach, Run, and Circle environments are adapted from Bullet-Safety (Gronauer, 2022), where a ball agent is used to perform different tasks. The Ant-vel, Swimmer-vel, and HalfCheetah-vel are adapted from Safety-Gym (Gronauer, 2022), where Ant, Swimmer, and HalfCheetah agents perform direction-based tasks under velocity constraints.

---

[6]https://github.com/sfujim/TD3, https://github.com/sfujim/TD3_BC
[7]https://github.com/liuzuxin/OSRL

*Table 3.* Some hyperparameters of our methods and baselines.

| Parameter | Setting |
|---|---|
| Latent variable dimension | 4 |
| Expectile for IQL | 0.9 for Reach, and 0.75 for all other environments |
| Bias regularizer $\lambda$ for CPL ($\alpha$) | 1.0 for Reach and Run, and 0.5 for all other environments |
| Bound for $\pi_h$ ($A_{\max}$) | 3.0 |
| Prior regularization coefficient $\beta_{\text{reg}}$ in Eqs. 11,12 | 1.0 in offline settings, and 0.01 in online settings |
| Behavior cloning weight in TD3+BC ($\beta_{\text{BC}}$) | Tuned over $\{1.0, 20.0, 200.0, 2000.0\}$ for each environment |
| Regularization coefficient for RC ($\kappa$) | 0.0005 |

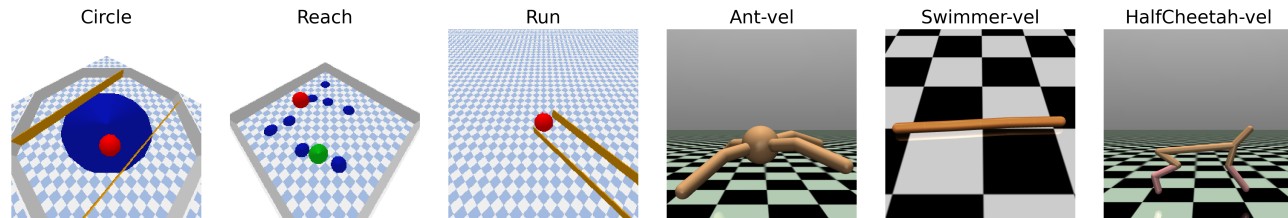

Circle   Reach   Run   Ant-vel   Swimmer-vel   HalfCheetah-vel

*Figure 5.* Visualization of the six downstream environments considered in this work. In the left three panels, the red dot denotes the agent; wall constraints are present in the first and third panels, while the blue region in the second panel indicates a risky area.

**Reach**   At the beginning of each episode, six hazardous regions are randomly generated. The agent is required to navigate to a target location while avoiding hazardous regions. The safety reward and a set of goal-specific rewards are defined as

$$r_{\text{share}} = -K \cdot \mathbb{I}(\text{distance to any risk area} < \delta_1) \tag{24}$$

$$r_{\text{goal}} = \mathbb{I}(\|s - s_{\text{target}}\|_2^2 < \delta_2), \tag{25}$$

where $s$ denotes the agent's current position and $s_{\text{target}}$ denotes an target location associated with different goal-specific rewards. Notably, $s_{\text{target}}$ is not included in the agent's state representation; thus, the agent can only infer target information implicitly through the reward signal. During annotation, we consider the four corner locations of the square map as targets, and use the corresponding goal-specific rewards $r_{\text{goal}}$ to induce crowd variation, i.e., the user-specific component $r_{\text{user}}$. The final annotation reward is then given by the combination $r_{\text{user}} + r_{\text{shared}}$. Such an option is motivated by the observation that behaviors for reaching arbitrary locations in downstream tasks can be composed from those for reaching the corner targets. In the downstream stage, the shared safety reward $r_{\text{shared}}$ is not available; instead, we have access only to goal-specific rewards associated with arbitrary target locations, corresponding to a continuum of downstream tasks, and report performance averaged across these tasks.

**Run**   This environment contains two passable walls located at $y = 0.5$ and $y = -0.5$. The agent is required to move along the $x$-axis at a target speed $v_{\text{target}} \in [0, 5.0]$. The safety reward and a set of goal-specific rewards are defined as

$$r_{\text{share}} = -K \cdot \mathbb{I}(|s_y| > 0.5) \tag{26}$$

$$r_{\text{goal}} = \exp(-0.5|v_x - v_{\text{target}}|), \tag{27}$$

where $s$ and $v$ denote the agent's position and velocity, respectively. During annotation, preference labels are generated using annotation rewards composed of the shared safety reward and two target speeds, $v_{\text{target}} \in 0, 5.0$. In the downstream stage, we use goal-specific rewards with arbitrary target speeds $v_{\text{target}} \in [0, 5.0]$.

**Circle**   Similar to Run, the environment contains two passable walls located at $x = 6.0$ and $y = -6.0$. The agent is required to move at a target tangential speed $v_{\text{target}} \in [0, 6.0]$ while rotating around the origin $(0, 0)$. The safety reward and a set of goal-specific rewards are defined as

$$r_{\text{share}} = -K \cdot \mathbb{I}(|x| > 6.0) \tag{28}$$

$$r_{\text{goal}} = -\exp(-0.5|v_{\text{tangent}} - v_{\text{target}}|) \big/ \left(1 + \left|\|s\|_2^2 - 7.0\right|\right), \tag{29}$$

where $s$ denotes the agent's position and $v_{\text{tangent}}$ is the tangential velocity around the origin. During annotation, preference labels are generated using two target speeds, $v_{\text{target}} = 0$ and $v_{\text{target}} = 6.0$. In the downstream stage, we consider arbitrary target speed $v_{\text{target}} \in [0, 6.0]$.

**Ant-vel, Swimmer-vel, and HalfCheetah-vel** In these environments, Ant, Swimmer, and HalfCheetah agents are required to move as fast as possible along a target direction $u = (\cos\theta_{\text{target}}, \sin\theta_{\text{target}})$, while ensuring that the overall speed does not exceed a threshold $\delta_v$. The reward functions are defined as

$$r_{\text{share}} = -K \cdot \mathbb{I}(\|v\| > \delta_v) \tag{30}$$

$$r_{\text{goal}} = v_x u_x + v_y u_y, \tag{31}$$

where $v$ denotes the agent's velocity. For Ant and Swimmer, $\theta_{\text{target}} \in \{0, \frac{\pi}{2}, \pi, \frac{3\pi}{2}\}$ combined with $r_{\text{share}}$ are used for annotation, and goal-specific reward $r_{\text{goal}}$ with $\theta_{\text{target}} = i \cdot 2\pi/40$ for $i = 0, 1, \ldots, 39$ are used as downstream task rewards. Due to structural constraints, HalfCheetah can only move forward or backward, resulting in two feasible direction $\theta_{\text{target}} = 0$ and $\theta_{\text{target}} = \pi$, both of which are used for annotation and downstream training.

**Discussion** In the Reach and Run environments, the shared objective and task objective are largely non-conflicting, leading to similar task performance with or without safety constraints. In contrast, in Circle and velocity-based tasks, the shared and task objectives are inherently conflicting, resulting in a larger task performance gap between safe agents and unsafe agents. We consider environments where user-specific preferences are defined by target velocity, goal, or direction, and safety is implicitly encoded through region or velocity constraints, covering most standard Safe RL benchmarks. We focus on tasks of moderate difficulty, where all methods can achieve high task performance, enabling clear evaluation of safety. In more challenging tasks, where all methods struggle, it becomes difficult to isolate the impact of safety due to the lack of a reliable performance reference on task completion.

## C.2. Offline Data Generation

To construct the crowd preference datasets, we first collect trajectories to form an offline dataset $\mathcal{D}_\tau$. We train agents using SAC for each goal-specific reward. Training is conducted for a total of $N_\tau$ steps: during the first $N_\tau/2$ steps, the agent optimizes only the task reward $r_{\text{user}}$, while during the remaining $N_\tau/2$ steps it optimizes the full reward $r_{\text{share}} + r_{\text{user}}$. The resulting replay buffer of size $N_\tau$ constitutes the offline dataset $\mathcal{D}_\tau$. This procedure ensures that $\mathcal{D}_\tau$ contains a mixture of unsafe and safe trajectory from all goals.

Besides, for all environments, we use a penalty coefficient $K = 10.0$, which allows the agent to achieve good task performance while maintaining near-zero cost across tasks. Although using a larger $K$ is possible, we find that it significantly harms exploration.

## C.3. Preference Generation

In the main experiments, we adopt a *balanced preference dataset*, where each task contributes an equal amount of preference data. Specifically, we sample $N_{\text{pref}}$ trajectory-pair sets from the offline dataset $\mathcal{D}_\tau$. Each set contains $S$ trajectory segment pairs, with each segment having length $L$. For each of the $M$ goal-specific rewards used for annotation (corresponding to different user-specific rewards $r_{\text{user}}$), preference labels are generated for all trajectory sets using the combined reward $r_{\text{share}} + r_{\text{user}}$, resulting in a total of $N_{\text{pref}} \times S \times M$ preference sets.[8] Although the annotation reward underlying each preference set are unobserved, we know all preference labels within the same set are generated under the same annotation reward. In Section 6.2, we additionally consider *imbalanced preference settings*, where preferences associated with one goal-specific reward dominate the dataset: after sampling $N_{\text{pref}}$ trajectory-pair sets, all sets are annotated under a single major annotation reward, whereas each remaining reward annotates only $0.1N_{\text{pref}}$ sets.

In addition, we consider a regret-based preference model, which requires optimal advantages rather than rewards to generate preference labels. Following CPL (Hejna et al., 2024), we add $\mathcal{D}_\tau$ to the replay buffer and re-training SAC policy using the full reward $r_{\text{share}} + r_{\text{user}}$ until convergence. The value function during training is then used to compute optimal advantages for preference label generation.

---

[8]We adopt this dense annotation setting in the main experiments. In Section D.5, we further consider a sparse annotation setting where trajectory sets are generated independently for each annotation reward.

Table 4 summarizes the hyperparameters used for offline data generation and preference generation.

*Table 4.* Hyperparameters for Offline Data Generation and Preference Generation.

| Parameter | Setting |
|---|---|
| SAC steps ($N_\tau$) | 5M for Ant-vel, and 2M for others |
| Penalty coefficient ($K$) | 10.0 |
| Number of $r_{\text{user}}$ ($M$) | 2 for Run, Circle and HalfCheetah, 4 for others |
| Number of preference sets ($N_{\text{pref}}$) (balanced) | 5000 from each task |
| Number of preference sets ($N_{\text{pref}}$) (imbalanced) | 10000 from a major task, and 1000 from other tasks |
| Context set size ($S$) | 16 |
| Segment length ($L$) | 16 for Reach, and 64 for others. |

# D. Additional Experimental Results

## D.1. Performance under Online Downstream Settings

In this section, we evaluate our methods under the online downstream setting, where the high-level policy is trained according to Eq. 11 using samples collected through online interaction with the environment. Table 5 reports the quantitative task and cost performance of different methods, while Figure 6 visualizes their performance fronts. Overall, the observed results are consistent with those in the offline downstream setting. Finally, Figure 7 shows performance over the first 0.5M online interaction steps, demonstrating that our method achieves substantially higher sample efficiency compared to online training from scratch. We attribute this improvement to the reuse of preference-aligned skills learned offline, which allows the high-level policy to focus on composing existing behaviors rather than relearning low-level behaviors through online exploration. As a result, our approach reaches strong performance with significantly fewer online interactions, suggesting that our approach is well suited to offline-to-online RL scenarios with limited interaction budgets.

*Table 5.* Normalized performance under online downstream settings. $\overline{\text{RC}}$ is the RC's performance averaged on all $\omega \in \{0.1, 0.2, ..., 1.0\}$

| Environment | Stats | Oracle | Task-Only | SOPL | RC($\omega = 0.5$) | Safe-VPL | Safe-CPL |
|---|---|---|---|---|---|---|---|
| Reach | Norm Rew | 1.00 | $0.96 \pm .01$ | $0.76 \pm .01$ | $0.73 \pm .01$ | $0.91 \pm .01$ | $0.90 \pm .03$ |
| | Norm Cost | 0.033 | $1.000 \pm .02$ | $0.083 \pm .01$ | $0.081 \pm .02$ | $0.124 \pm .03$ | $0.074 \pm .07$ |
| Run | Norm Rew | 1.00 | $1.00 \pm .00$ | $1.00 \pm .00$ | $1.00 \pm .00$ | $1.00 \pm .00$ | $0.99 \pm .01$ |
| | Norm Cost | 0.000 | $1.000 \pm .50$ | $0.000 \pm .00$ | $0.000 \pm .00$ | $0.000 \pm .00$ | $0.000 \pm .00$ |
| Circle | Norm Rew | 1.00 | $1.28 \pm .01$ | $1.08 \pm .01$ | $1.07 \pm .00$ | $0.97 \pm .03$ | $0.94 \pm .08$ |
| | Norm Cost | 0.000 | $1.000 \pm .02$ | $0.000 \pm .00$ | $0.000 \pm .00$ | $0.000 \pm .00$ | $0.000 \pm .00$ |
| Ant-vel | Norm Rew | 1.00 | $1.11 \pm .11$ | $0.84 \pm .07$ | $0.90 \pm .01$ | $0.94 \pm .03$ | $0.88 \pm .07$ |
| | Norm Cost | 0.000 | $1.000 \pm .00$ | $0.039 \pm .02$ | $0.100 \pm .14$ | $0.001 \pm .00$ | $0.028 \pm .01$ |
| Swimmer-vel | Norm Rew | 1.00 | $1.65 \pm .18$ | $1.52 \pm .09$ | $0.81 \pm .07$ | $0.92 \pm .03$ | $1.00 \pm .13$ |
| | Norm Cost | 0.000 | $1.000 \pm .17$ | $0.009 \pm .01$ | $0.000 \pm .00$ | $0.001 \pm .00$ | $0.022 \pm .02$ |
| HalfCheetah-vel | Norm Rew | 1.00 | $2.80 \pm .29$ | $0.90 \pm .07$ | $0.82 \pm .10$ | $0.96 \pm .02$ | $0.89 \pm .04$ |
| | Norm Cost | 0.000 | $1.000 \pm .00$ | $0.007 \pm .01$ | $0.076 \pm .06$ | $0.004 \pm .00$ | $0.003 \pm .00$ |
| Average | Norm Rew | 1.00 | 1.47 | 0.95 | 0.82 | 0.95 | 0.93 |
| | Norm Cost | 0.01 | 1.00 | 0.02 | 0.04 | 0.02 | 0.02 |

## D.2. Raw Data in Main Comparison

We present the raw performance data under the offline and online downstream settings in Table 6 and Table 7, respectively. The results also include the random policy, which is used to compute the normalized reward and cost.

## D.3. Ablation Results on Additional Environments

This section presents ablation results on the regularization weight $\beta_{\text{reg}}$, preference noise ratios, and crowd size across additional environments; see Figures 8, 9 and 10, which is consistent with the conclusions in Section 6.

*Table 6.* Raw task and safety performance under offline downstream settings

| Env | Stats | Random | CDT | Task-Only | SOPL | RC($\omega = 0.5$) | Safe-VPL | Safe-CPL |
|---|---|---|---|---|---|---|---|---|
| Reach | R | -4.5 | 18.8 | $19.7 \pm .1$ | $18.3 \pm .6$ | $14.8 \pm .2$ | $18.3 \pm .8$ | $18.2 \pm 1.0$ |
| | C | 7.0 | 0.1 | $2.2 \pm .0$ | $0.1 \pm .0$ | $0.2 \pm .0$ | $0.4 \pm .1$ | $0.2 \pm .1$ |
| Run | R | -151.7 | 94.7 | $93.7 \pm .1$ | $91.8 \pm .4$ | $93.5 \pm .7$ | $82.1 \pm 2.1$ | $86.1 \pm 1.7$ |
| | C | 45.6 | 0.0 | $67.4 \pm 5.5$ | $0.0 \pm .0$ | $0.0 \pm .0$ | $0.0 \pm .0$ | $0.0 \pm .0$ |
| Circle | R | 19.6 | 149.8 | $185.5 \pm 2.4$ | $155.7 \pm 1.7$ | $161.1 \pm .5$ | $136.9 \pm 1.0$ | $137.6 \pm 5.3$ |
| | C | 63.1 | 0.0 | $63.0 \pm 1.2$ | $0.0 \pm .0$ | $0.0 \pm .0$ | $0.0 \pm .0$ | $0.0 \pm .1$ |
| Ant-vel | R | -65.7 | 2306.8 | $2860.7 \pm 42.3$ | $2259.4 \pm 27.3$ | $1753.4 \pm 466.3$ | $2058.0 \pm 36.1$ | $1746.9 \pm 228.2$ |
| | C | 0.7 | 0.8 | $488.1 \pm 3.4$ | $3.3 \pm 1.2$ | $38.2 \pm 58.7$ | $0.6 \pm .1$ | $3.5 \pm 1.9$ |
| Swimmer-vel | R | 1.5 | 42.7 | $100.0 \pm 1.5$ | $56.6 \pm 1.4$ | $35.8 \pm 3.1$ | $37.7 \pm .3$ | $42.4 \pm 1.5$ |
| | C | 63.2 | 0.0 | $27.4 \pm 2.4$ | $0.4 \pm .2$ | $0.0 \pm .0$ | $0.0 \pm .0$ | $0.1 \pm .1$ |
| HalfCheetah-vel | R | -210.2 | 2688.7 | $5166.3 \pm 1053.2$ | $2477.3 \pm .0$ | $1060.1 \pm 317.0$ | $2581.5 \pm 59.6$ | $2452.5 \pm 6.5$ |
| | C | 0.1 | 0.0 | $696.3 \pm 134.9$ | $9.5 \pm .0$ | $74.2 \pm 57.4$ | $2.7 \pm 1.6$ | $12.3 \pm 12.2$ |

*Table 7.* Raw task and safety performance under online downstream settings

| Env | Stats | Random | SAC | Task-Only | SOPL | RC($\omega = 0.5$) | Safe-VPL | Safe-CPL |
|---|---|---|---|---|---|---|---|---|
| Reach | R | -4.5 | 21.8 | $20.7 \pm .1$ | $15.4 \pm .2$ | $14.8 \pm .2$ | $19.6 \pm .2$ | $19.1 \pm .8$ |
| | C | 7.0 | 0.1 | $2.7 \pm .0$ | $0.2 \pm .0$ | $0.2 \pm .0$ | $0.3 \pm .1$ | $0.2 \pm .2$ |
| Run | R | -151.7 | 93.9 | $94.5 \pm .0$ | $94.3 \pm .1$ | $93.5 \pm .7$ | $94.0 \pm .6$ | $91.5 \pm 1.8$ |
| | C | 45.6 | 0.0 | $26.0 \pm 12.9$ | $0.0 \pm .0$ | $0.0 \pm .0$ | $0.0 \pm .0$ | $0.0 \pm .0$ |
| Circle | R | 19.6 | 151.4 | $188.3 \pm .6$ | $162.3 \pm 1.4$ | $161.1 \pm .5$ | $147.3 \pm 3.9$ | $143.7 \pm 1.8$ |
| | C | 63.1 | 0.0 | $66.1 \pm 1.1$ | $0.0 \pm .0$ | $0.0 \pm .0$ | $0.0 \pm .0$ | $0.0 \pm .0$ |
| Ant-vel | R | -65.7 | 2317.5 | $2590.1 \pm 264.0$ | $1938.5 \pm 175.5$ | $2091.0 \pm 33.5$ | $2171.7 \pm 61.4$ | $2023.3 \pm 171.9$ |
| | C | 0.7 | 0.2 | $523.2 \pm .2$ | $20.4 \pm 9.8$ | $52.3 \pm 71.1$ | $0.6 \pm .1$ | $14.9 \pm 7.4$ |
| Swimmer-vel | R | 1.5 | 43.8 | $71.3 \pm 7.8$ | $65.8 \pm 4.0$ | $35.8 \pm 3.1$ | $40.4 \pm 1.3$ | $43.9 \pm 5.4$ |
| | C | 63.2 | 0.0 | $21.9 \pm 3.7$ | $0.2 \pm .2$ | $0.0 \pm .0$ | $0.0 \pm .0$ | $0.5 \pm .5$ |
| HalfCheetah-vel | R | -210.2 | 2806.3 | $8245.2 \pm 889.3$ | $2525.7 \pm 224.7$ | $2301.2 \pm 317.0$ | $2687.8 \pm 6.9$ | $2465.7 \pm 129.6$ |
| | C | 0.1 | 0.0 | $980.8 \pm .6$ | $7.3 \pm 7.3$ | $74.2 \pm 57.4$ | $3.6 \pm 2.1$ | $2.5 \pm 2.4$ |

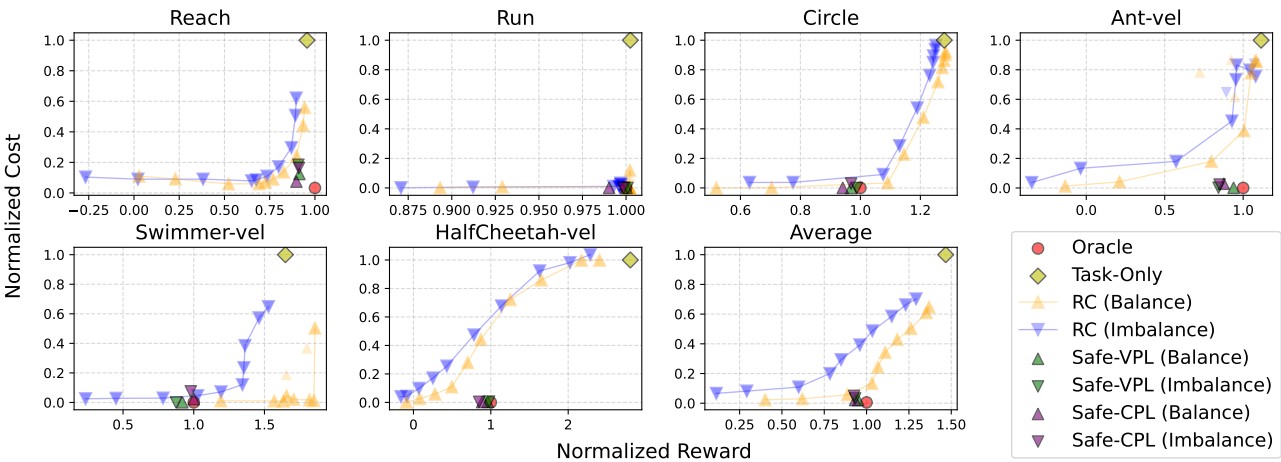

*Figure 6.* Performance fronts of different algorithms and RC with various trade-off weights $\omega$ using balanced and imbalanced crowd preference dataset under the online downstream setting.

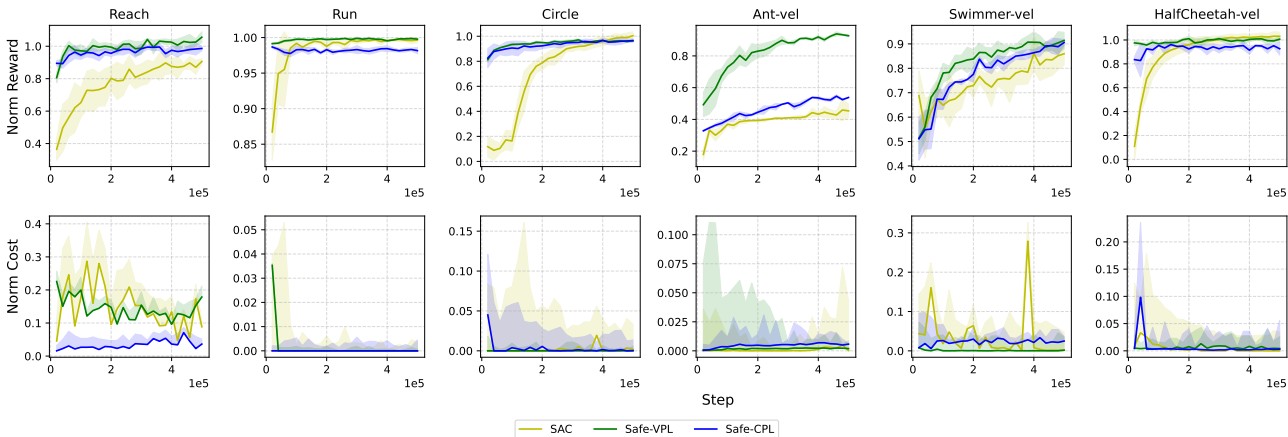

*Figure 7.* Performance curves of different algorithms during the first 0.5M training steps under online downstream settings.

### D.4. Ablations on Additional Settings

**Number of preference sets $N_{\text{pref}}$ and size of each preference set $\mathcal{S}_z$.** We further analyze the sensitivity of our method to the number of preference sets $N_{\text{pref}}$ and the size of each preference set $\mathcal{S}_z$. As shown in Figure 11, reducing the number of preference sets (i.e., decreasing the overall dataset size) primarily degrades safety performance, while only mildly affecting task performance. In contrast, Figure 12 studies the effect of reducing the size of each preference set $\mathcal{S}_z$ while keeping the total number of samples $N_{\text{pref}} \times |\mathcal{S}_z|$ fixed. In this setting, compared with task performance, safety performance degrades significantly. This is because a smaller $\mathcal{S}_z$ limits the model's ability to capture user-level behavioral patterns through the latent variable $z$, causing it to instead encode individual trajectory-level preferences. Specifically, when comparing two unsafe trajectories, the less unsafe one may be incorrectly treated as preferred within a given context $z$. As $\mathcal{S}_z$ increases, each trajectory is more likely to be compared against safe trajectories under the same context, allowing the model to correctly identify and exclude such less unsafe behaviors from the learned low-level skills.

**Training steps.** We further study the effect of low-level policy training steps on downstream performance. By varying the number of training steps, we explicitly control the quality of the learned skills. Figure 13 shows as the number of training steps decreases, task performance degrades slightly, reflecting reduced skill optimality. In contrast, safety performance remains relatively stable, indicating that the shared safety structure is more robust to imperfect skill learning. This suggests that safety properties can be largely preserved even when the learned low-level policies are suboptimal.

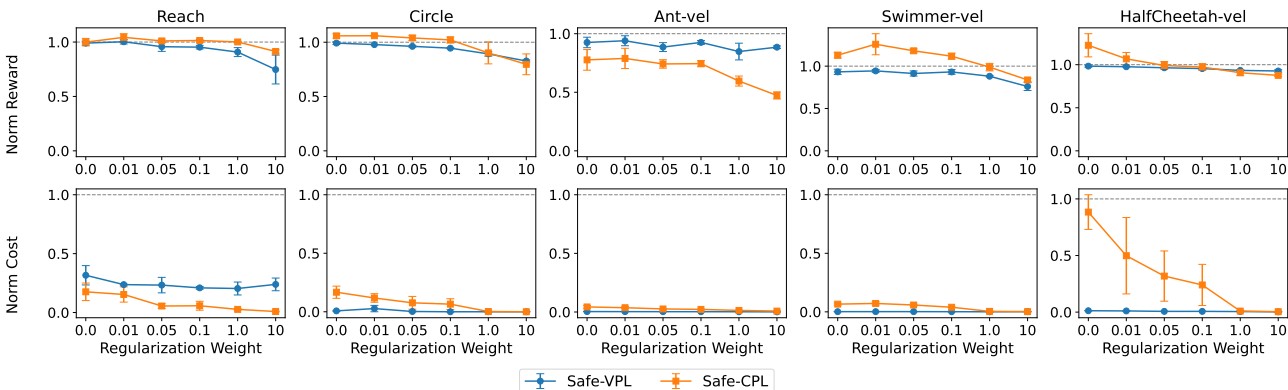

*Figure 8.* Performance of our methods under different prior regularization weights $\beta_{\mathrm{reg}} \in [0.0, 0.01, 0.05, 0.1, 1.0, 10.0]$ in Eq. 11.

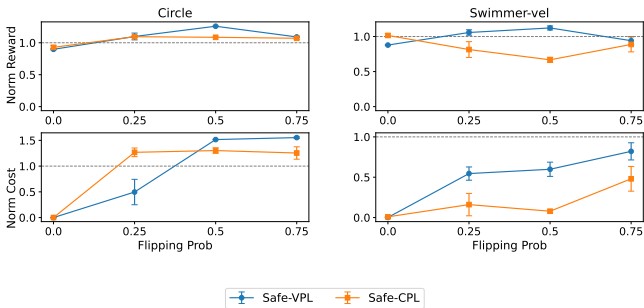

*Figure 9.* Performance of our methods under different preference noise ratios.

### D.5. Dense and Sparse Annotation for Crowd Preference Data

In the main experiments, we adopt a *dense annotation* setting, where the same $N_{\mathrm{pref}}$ trajectory sets is labeled under all $M$ annotation tasks, resulting in $N_{\mathrm{pref}} \times M$ preference sets in total. In this subsection, we additionally consider a *sparse annotation* setting, where for each annotation reward, we independently samples and labels $N_{\mathrm{pref}}$ trajectory sets, with no trajectory pairs shared across different users, while keeping the total number of preference pairs the same. A comparison between the two settings is shown in Table 8, from which we observe that the choice of annotation scheme has no significant impact on performance, except for a slight degradation in safety performance for CPL..

### D.6. Safe-VPL with Full Dataset and Preference-only Dataset

Following the original VPL formulation (Poddar et al., 2024), Safe-VPL requires an additional offline dataset $\mathcal{D}_{\tau}$ in Eq. 9 during skill discovery, whereas Safe-CPL directly learns policies from the preference dataset without $\mathcal{D}_{\tau}$. As a result, Safe-VPL benefits from substantially more offline data, making a direct comparison between the two methods inherently unfair. To address this issue, we construct a variant denoted as **Safe-VPL (Pref only)**, in which the preference dataset $\mathcal{D}_{\mathrm{pref}}$ is reorganized into an offline transition dataset and used as a substitute for $\mathcal{D}_{\tau}$. Offline RL is then performed on this reorganized dataset, thereby ensuring that both algorithms have access to the same amount of training data. As shown in Table 9, when the data size is the same, the performance of Safe-VPL degrades, suggesting that access to additional offline data can benefit skill discovery, and the choice between the two methods can be adapted based on data availability.

*Table 8.* Performance comparison of our method with dense-annotated and sparse-annotated preference data.

| Environment | Stats | Dense | | Sparse | |
|---|---|---|---|---|---|
| | | Safe-VPL | Safe-CPL | Safe-VPL | Safe-CPL |
| Reach | Norm Rew | $1.00 \pm .03$ | $1.00 \pm .01$ | $0.97 \pm .02$ | $1.03 \pm .01$ |
| | Norm Cost | $0.141 \pm .02$ | $0.059 \pm .01$ | $0.146 \pm .04$ | $0.249 \pm .02$ |
| Run | Norm Rew | $0.99 \pm .00$ | $0.96 \pm .01$ | $0.99 \pm .00$ | $0.97 \pm .01$ |
| | Norm Cost | $0.000 \pm .00$ | $0.000 \pm .00$ | $0.000 \pm .00$ | $0.000 \pm .00$ |
| Circle | Norm Rew | $0.90 \pm .01$ | $0.93 \pm .03$ | $0.92 \pm .01$ | $0.92 \pm .01$ |
| | Norm Cost | $0.000 \pm .00$ | $0.000 \pm .00$ | $0.000 \pm .00$ | $0.099 \pm .06$ |
| Ant-vel | Norm Rew | $0.91 \pm .01$ | $0.74 \pm .10$ | $0.91 \pm .01$ | $0.62 \pm .03$ |
| | Norm Cost | $0.002 \pm .00$ | $0.008 \pm .00$ | $0.002 \pm .00$ | $0.020 \pm .01$ |
| Swimmer-vel | Norm Rew | $0.88 \pm .01$ | $1.01 \pm .03$ | $0.90 \pm .02$ | $1.17 \pm .04$ |
| | Norm Cost | $0.001 \pm .00$ | $0.008 \pm .01$ | $0.001 \pm .00$ | $0.120 \pm .03$ |
| HalfCheetah-vel | Norm Rew | $0.96 \pm .02$ | $0.91 \pm .01$ | $0.99 \pm .01$ | $0.87 \pm .04$ |
| | Norm Cost | $0.005 \pm .00$ | $0.026 \pm .03$ | $0.007 \pm .00$ | $0.101 \pm .10$ |
| Average | Norm Rew | 0.94 | 0.92 | 0.95 | 0.93 |
| | Norm Cost | 0.02 | 0.02 | 0.03 | 0.10 |

*Table 9.* Comparison between Safe-VPL, Safe-VPL (Pref only), and Safe-CPL under equal data size

| Environment | Stats | Safe-VPL | Safe-VPL (Pref only) | Safe-CPL |
|---|---|---|---|---|
| Reach | Norm Rew | $1.00 \pm .03$ | $0.94 \pm .04$ | $1.00 \pm .01$ |
| | Norm Cost | $0.141 \pm .02$ | $0.234 \pm .10$ | $0.059 \pm .01$ |
| Run | Norm Rew | $0.99 \pm .00$ | $0.99 \pm .00$ | $0.96 \pm .01$ |
| | Norm Cost | $0.000 \pm .00$ | $0.000 \pm .00$ | $0.000 \pm .00$ |
| Circle | Norm Rew | $0.90 \pm .01$ | $0.89 \pm .00$ | $0.93 \pm .03$ |
| | Norm Cost | $0.000 \pm .00$ | $0.000 \pm .00$ | $0.000 \pm .00$ |
| Ant-vel | Norm Rew | $0.91 \pm .01$ | $0.82 \pm .01$ | $0.74 \pm .10$ |
| | Norm Cost | $0.002 \pm .00$ | $0.002 \pm .00$ | $0.008 \pm .00$ |
| Swimmer-vel | Norm Rew | $0.88 \pm .01$ | $0.79 \pm .14$ | $1.01 \pm .03$ |
| | Norm Cost | $0.001 \pm .00$ | $0.000 \pm .00$ | $0.008 \pm .01$ |
| HalfCheetah-vel | Norm Rew | $0.96 \pm .02$ | $0.89 \pm .06$ | $0.91 \pm .01$ |
| | Norm Cost | $0.005 \pm .00$ | $0.001 \pm .00$ | $0.026 \pm .03$ |
| Average | Norm Rew | 0.94 | 0.88 | 0.92 |
| | Norm Cost | 0.02 | 0.04 | 0.02 |

*Table 10.* Offline downstream performance comparison under balanced and imbalanced preference settings. $\overline{\mathrm{RC}}$ is the RC's performance averaged on all $\omega \in \{0.1, 0.2, ..., 1.0\}$

| Environment | Stats | Balance | | | Imbalance | | |
|---|---|---|---|---|---|---|---|
| | | $\overline{\mathrm{RC}}$ | Safe-VPL | Safe-CPL | $\overline{\mathrm{RC}}$ | Safe-VPL | Safe-CPL |
| Reach | Norm Rew | $0.88 \pm .11$ | $1.00 \pm .03$ | $1.00 \pm .01$ | $1.04 \pm .00$ | $0.97 \pm .01$ | $0.98 \pm .01$ |
| | Norm Cost | $0.299 \pm .15$ | $0.141 \pm .02$ | $0.059 \pm .01$ | $0.434 \pm .06$ | $0.182 \pm .02$ | $0.067 \pm .05$ |
| Run | Norm Rew | $0.99 \pm .00$ | $0.99 \pm .00$ | $0.96 \pm .01$ | $0.91 \pm .01$ | $0.98 \pm .01$ | $0.97 \pm .01$ |
| | Norm Cost | $0.030 \pm .03$ | $0.000 \pm .00$ | $0.000 \pm .00$ | $0.473 \pm .47$ | $0.000 \pm .00$ | $0.000 \pm .00$ |
| Circle | Norm Rew | $0.96 \pm .01$ | $0.90 \pm .01$ | $0.93 \pm .03$ | $0.82 \pm .03$ | $0.87 \pm .03$ | $0.92 \pm .05$ |
| | Norm Cost | $0.128 \pm .01$ | $0.000 \pm .00$ | $0.000 \pm .00$ | $0.144 \pm .04$ | $0.000 \pm .00$ | $0.000 \pm .00$ |
| Ant-vel | Norm Rew | $0.90 \pm .03$ | $0.91 \pm .01$ | $0.74 \pm .10$ | $0.85 \pm .08$ | $0.86 \pm .04$ | $0.73 \pm .10$ |
| | Norm Cost | $0.371 \pm .08$ | $0.002 \pm .00$ | $0.008 \pm .00$ | $0.335 \pm .02$ | $0.002 \pm .00$ | $0.018 \pm .01$ |
| Swimmer-vel | Norm Rew | $1.29 \pm .08$ | $0.88 \pm .01$ | $1.01 \pm .03$ | $0.61 \pm .07$ | $0.85 \pm .05$ | $1.02 \pm .00$ |
| | Norm Cost | $0.121 \pm .01$ | $0.001 \pm .00$ | $0.008 \pm .01$ | $0.137 \pm .01$ | $0.000 \pm .00$ | $0.032 \pm .02$ |
| HalfCheetah-vel | Norm Rew | $1.41 \pm .34$ | $0.96 \pm .02$ | $0.91 \pm .01$ | $1.35 \pm .18$ | $0.97 \pm .01$ | $0.84 \pm .05$ |
| | Norm Cost | $1.169 \pm .14$ | $0.005 \pm .00$ | $0.026 \pm .03$ | $1.190 \pm .20$ | $0.010 \pm .00$ | $0.136 \pm .13$ |
| Average | Norm Rew | $1.07$ | $0.94$ | $0.92$ | $0.93$ | $0.92$ | $0.91$ |
| | Norm Cost | $0.35$ | $0.02$ | $0.02$ | $0.45$ | $0.03$ | $0.04$ |

*Table 11.* Online downstream performance comparison under balanced and imbalanced preference settings. $\overline{\mathrm{RC}}$ is the RC's performance averaged on all $\omega \in \{0.1, 0.2, ..., 0.9\}$

| Environment | Stats | Balance | | | Imbalance | | |
|---|---|---|---|---|---|---|---|
| | | $\overline{\mathrm{RC}}$ | Safe-VPL | Safe-CPL | $\overline{\mathrm{RC}}$ | Safe-VPL | Safe-CPL |
| Reach | Norm Rew | $0.53 \pm .12$ | $0.91 \pm .01$ | $0.88 \pm .01$ | $0.53 \pm .11$ | $0.91 \pm .01$ | $0.91 \pm .03$ |
| | Norm Cost | $0.230 \pm .03$ | $0.119 \pm .04$ | $0.041 \pm .01$ | $0.191 \pm .07$ | $0.170 \pm .04$ | $0.112 \pm .09$ |
| Run | Norm Rew | $0.97 \pm .01$ | $1.00 \pm .00$ | $0.99 \pm .01$ | $0.78 \pm .16$ | $1.00 \pm .00$ | $1.00 \pm .00$ |
| | Norm Cost | $0.010 \pm .01$ | $0.000 \pm .00$ | $0.000 \pm .00$ | $0.009 \pm .01$ | $0.000 \pm .00$ | $0.000 \pm .00$ |
| Circle | Norm Rew | $0.96 \pm .07$ | $0.98 \pm .03$ | $0.92 \pm .08$ | $0.44 \pm .58$ | $1.00 \pm .00$ | $0.94 \pm .10$ |
| | Norm Cost | $0.211 \pm .14$ | $0.000 \pm .00$ | $0.000 \pm .00$ | $0.175 \pm .17$ | $0.000 \pm .00$ | $0.000 \pm .00$ |
| Ant-vel | Norm Rew | $0.92 \pm .09$ | $0.93 \pm .03$ | $0.82 \pm .01$ | $0.80 \pm .21$ | $0.81 \pm .06$ | $0.86 \pm .02$ |
| | Norm Cost | $0.696 \pm .29$ | $0.001 \pm .00$ | $0.018 \pm .01$ | $0.695 \pm .30$ | $0.002 \pm .00$ | $0.019 \pm .01$ |
| Swimmer-vel | Norm Rew | $1.27 \pm .20$ | $0.91 \pm .03$ | $1.03 \pm .13$ | $0.34 \pm .01$ | $0.88 \pm .01$ | $0.95 \pm .00$ |
| | Norm Cost | $0.058 \pm .05$ | $0.000 \pm .00$ | $0.030 \pm .03$ | $0.009 \pm .01$ | $0.000 \pm .00$ | $0.043 \pm .00$ |
| HalfCheetah-vel | Norm Rew | $2.27 \pm .17$ | $0.95 \pm .02$ | $0.89 \pm .05$ | $2.10 \pm .07$ | $0.97 \pm .01$ | $0.85 \pm .00$ |
| | Norm Cost | $0.847 \pm .01$ | $0.003 \pm .00$ | $0.001 \pm .00$ | $0.845 \pm .01$ | $0.003 \pm .00$ | $0.002 \pm .00$ |
| Average | Norm Rew | $1.15$ | $0.95$ | $0.92$ | $0.83$ | $0.93$ | $0.92$ |
| | Norm Cost | $0.34$ | $0.02$ | $0.02$ | $0.32$ | $0.03$ | $0.03$ |

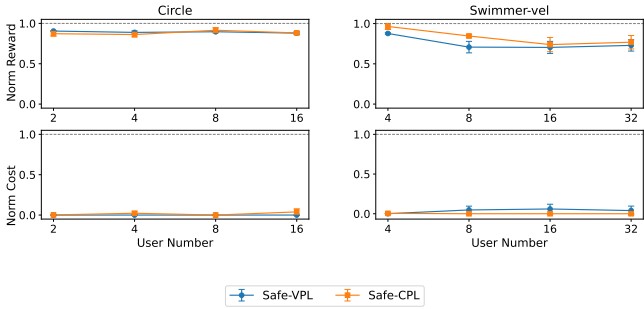

*Figure 10.* Performance of our methods under different crowd sizes.

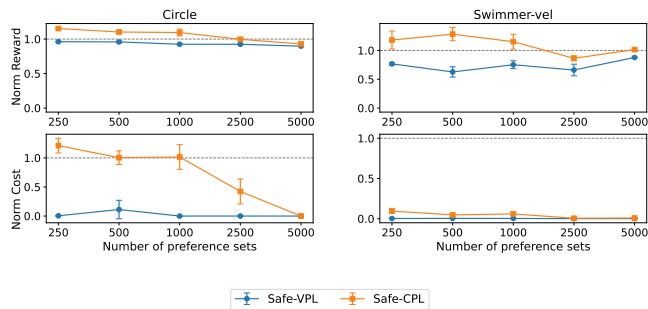

*Figure 11.* Performance of our methods under different number of preference sets $N_{\text{pref}}$.

# E. LLM Evaluation Details

**Task Description**  The environment consists of one query, *"Please talk about one kind of pets"*, together with three categories of responses: bird-related, dog-related, and cat-related responses. Each category contains both safe responses and unsafe responses. Safe responses consist only of normal pet-related descriptions (e.g., "Dogs are our friends"), while unsafe responses are created by appending toxic or offensive statements to otherwise normal pet-related descriptions (e.g., "Dogs are our friends. [harmful statement encouraging animal abuse]").

The task can be formulated as a single-step MDP, where each state is a pair $s = (y_1, y_2)$ of candidate responses and the action space is $a \in \{0, 1\}$, indicating whether the agent selects $y_1$ or $y_2$. The underlying reward is decomposed as

$$r(s, a, z) = r_{\text{user}}(s, a, z) + r_{\text{share}}(s, a).$$

Here, $r_{\text{user}}(s, a, z)$ is a user-specific reward induced by a preference ranking over response categories (e.g., $A > B > C$), assigning reward 1 if the selected response matches the user ranking and 0 otherwise. The shared component $r_{\text{share}}(s, a)$ captures the implicit safety objective of preferring safe responses over unsafe ones:

$$r_{\text{share}}(s, a) = \begin{cases} -K, & \text{if the selected response contains harmful content,} \\ 0, & \text{otherwise,} \end{cases}$$

where $K \gg 1$. As in the main paper, neither reward component is directly observable and both are only reflected implicitly through crowd preference data.

**Data Generation**  We construct a synthetic pet-safety dataset with imbalanced crowd preferences. Each preference pair is generated by sampling two safe responses from different response categories and injecting harmful content into one response with probability $0.5$. The final dataset contains 5000 preference pairs. To generate preference labels, we consider two users. User 1 contributes $80\%$ of the data and follows the ranking $A > B > C$, while User 2 contributes the remaining $20\%$ and follows $C > B > A$. Both users consistently prefer safe responses over harmful ones regardless of response category.

**Downstream Tasks**  We consider two downstream tasks with new rankings $B > A > C$ and $C > A > B$. These rankings do not match the preference ordering of either crowd user, meaning that simply following the majority preference

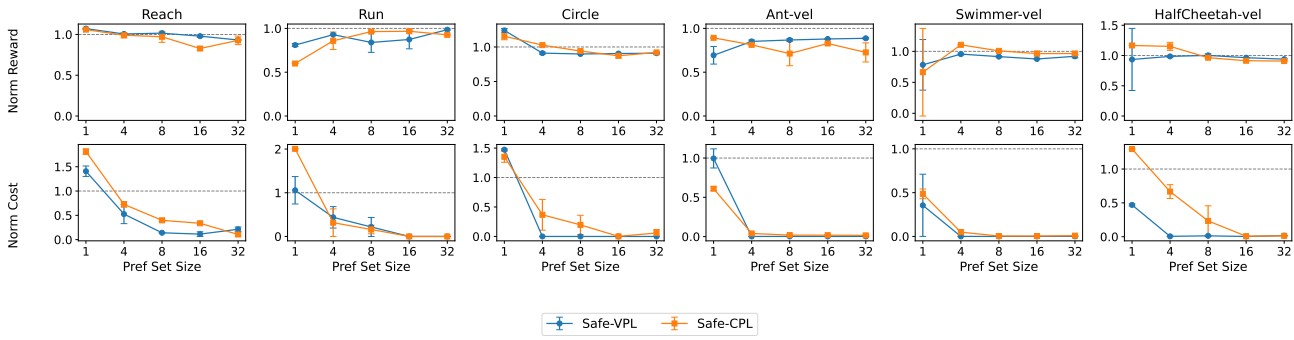

*Figure 12.* Performance of our methods under different size of the preference set $\mathcal{S}_z$.

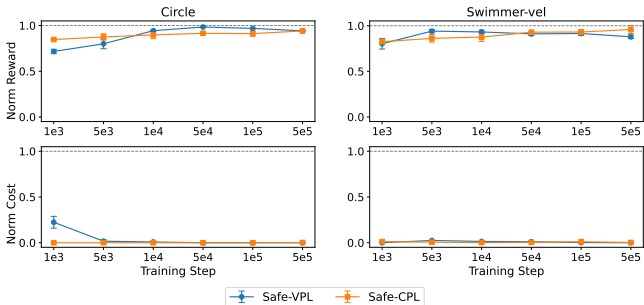

*Figure 13.* Performance of our methods under different low-level policy training steps.

or imitating a single user cannot maximize downstream reward. Instead, successful performance requires composing crowd preferences and generalizing beyond those directly observed in the crowd data.

Importantly, downstream training only provides the safety-agnostic reward $r_{\text{new}}$, without access to the shared safety objective. As a result, directly maximizing $r_{\text{new}}$ can encourage the selection of harmful responses whenever they belong to highly ranked categories.

**Evaluation** The test set contains all three category combinations among $A$, $B$, and $C$, where half of the pairs consist of two safe responses (Safe–Safe) and the other half consist of one safe and one unsafe response (Safe–Unsafe). We define the downstream task reward $r_{\text{new}}$ as $1$ if the selected response matches the task ranking and $0$ otherwise. The safety cost is defined as $1$ if the selected response contains harmful content and $0$ otherwise. Finally, we report the average downstream task reward and safety cost over all test samples.

**Implementation** **Task-only** trains a policy that takes the embeddings of two candidate responses as input and outputs a Bernoulli distribution over actions, optimized via REINFORCE using only $r_{\text{new}}$. **RC** first learns a unimodal reward model $r_{\text{uni}}$ from crowd preference data, then optimizes $(1 - \omega)r_{\text{new}} + \omega r_{\text{uni}}$ using the same policy architecture as Task-only. **Our method** learns a latent-conditioned reward $r(y, z)$ from crowd preferences and trains a high-level policy $\pi_h(z \mid s)$ via REINFORCE to adaptively select the latent reward used for response comparison. Specifically, the action distribution is produced through

$$p(a = 1) = \sigma(r(y_1, z) - r(y_2, z)), \qquad z = \pi_h(z \mid s).$$

# F. Extension to General Safety Constraints

Our work primarily focuses on hard constraint violations, as they provide a clear and verifiable definition of shared safety consensus across users, which is particularly important given the limited prior work and the inherent difficulty of defining shared structure from preferences. Specifically, our formulation relies on two key assumptions: (1) a sufficiently large safety penalty $K$, and (2) a binary cost function defined as $c(s, a) = -r_{\text{share}}(s, a) = K$ if $(s, a)$ is unsafe, and $c(s, a) = 0$ otherwise. Under these assumptions, the shared safety consensus can be formalized as:

**Definition (Hard-constraint safety consensus).** Avoid unsafe state–action pairs, i.e., $c(s_t, a_t) = 0$ for all $t$.

While this setting provides a clean starting point, we next discuss how our framework can be extended to more general forms of safety constraints.

## F.1. General Cost Functions

We first consider the case where the cost function is non-binary and varies across state–action pairs, i.e., $c(s, a) \geq 0$. In this setting, state–action pairs with sufficiently large penalties (e.g., exceeding a threshold as in Theorem 1) are consistently dispreferred by all users. This induces the following relaxed notion of safety consensus:

**Definition (High-cost avoidance consensus).** Avoid highly unsafe state–action pairs, i.e., $c(s_t, a_t) = 0$ for all $t$ and all $(s_t, a_t)$ such that $c(s_t, a_t) > K'$, where $K'$ is a sufficiently large constant.

Our proposed method naturally satisfies this form of consensus, as highly penalized behaviors are consistently filtered out during skill discovery.

## F.2. Threshold-based Safety Constraints

We next consider a more general setting where safety is expressed via cumulative cost constraints. In this case, each user $z$ generates preference labels according to the utility:

$$\sum_t r_{\text{user}}(s_t, a_t, z) + \sum_t r_{\text{share}}(s_t, a_t) = \sum_t r_{\text{user}}(s_t, a_t, z) - \sum_t c(s_t, a_t).$$

Under a Lagrangian formulation, the policy aligned with user $z$'s preferences can be characterized as the solution to the following constrained optimization problem:

$$\max_\pi \, \mathbb{E}_\pi \left[ \sum_t r_{\text{user}}(s_t, a_t, z) \right] \quad \text{s.t.} \quad \mathbb{E}_\pi \left[ \sum_t c(s_t, a_t) \right] \leq \zeta_z,$$

where $\zeta_z$ is a user-specific threshold on cumulative cost.

In this framework, users share the same cost function but differ in their tolerance levels. Since all users disprefer trajectories that violate the most permissive constraint, this induces a generalized notion of safety consensus:

**Definition (Threshold-based safety consensus).** Satisfy the most permissive constraint, i.e., $\sum_t c(s_t, a_t) \leq \max_z \zeta_z$.

This formulation generalizes the previous settings. In particular, the hard-constraint and high-cost avoidance consensus can be viewed as special cases where all or a subset of unsafe state–action pairs incur sufficiently large penalties, leading all users to effectively enforce a zero-tolerance constraint on these pairs.

## F.3. Approximate Satisfaction via Extended Skill Duration

Unlike prior skill learning methods that adopt fixed switching intervals (Ajay et al., 2020) or explicitly learn switching policies (Shankar & Gupta, 2020), our implementation allows the high-level policy to select a skill at every time step. This design is sufficient to satisfy the hard-constraint safety consensus: if each low-level policy avoids unsafe state–action pairs, any per-step composition preserves safety. However, this property does not extend to satisfying the threshold-based safety consensus. For example, consider a two-step MDP with two low-level policies: one incurs a cost of 1 at the first step, and the other incurs a cost of 1 at the second step. While each policy individually satisfies $\sum_t c_t \leq 1$, alternating between them results in a cumulative cost of 2, violating the constraint.

A simple solution is to increase the skill duration by fixing a switching interval $T'$. In the extreme case of $T' = T$ (episode-level switching), the threshold-based safety consensus is satisfied, but at the cost of reduced expressiveness. More generally, varying $T' \in [1, T]$ induces a trade-off between safety constraint satisfaction and task performance.

### F.4. Adapting the Framework to Longer Switching Intervals

Under the online downstream setting, our method can be naturally extended to a fixed switching interval $T'$. The high-level policy samples a latent variable $z_i'$ every $T'$ steps based on state $s_{i \cdot T'}$. Instead of storing primitive transitions $(s_t, a_t, r_t, s_{t+1})$, we construct a replay buffer containing aggregated transitions of the form $(s_{i \cdot T'}, z_i', r_i = \sum_{t=i \cdot T'}^{i \cdot T' + T'} r_t, s_{i \cdot T' + T'})$. which are used to estimate $Q(s_{i \cdot T'}, z_i')$ and update $\pi(z_i' \mid s_{i \cdot T'})$.

Extending this approach to the offline downstream setting is less straightforward, as the latent variable $z'$ associated with each trajectory segment is not observed, and thus the original dataset $\mathcal{D}_\tau = \{(s_t, a_t, r_t, s_{t+1})\}$ cannot be directly transformed into the aggregated form $\mathcal{D}_\tau'$. One possible solution is to train an auxiliary encoder $\tilde{q}(z' \mid \tau)$ that maps trajectory segments to latent variables by solving

$$\max_{\tilde{q}} \ \mathbb{E}_{\tau = \{s_t, a_t, \ldots, s_{t+k}, a_{t+k}\} \sim \mathcal{D}_\tau, \ z' \sim \tilde{q}(z'|\tau)} \left[ \sum_{j=t}^{t+k} \log \pi_l(a_j \mid s_j, z') \right],$$

with the low-level policy $\pi_l$ fixed. By segmenting $\mathcal{D}_\tau$ into length-$T'$ trajectories and inferring the corresponding $z_i'$ via $\tilde{q}(z' \mid \tau)$, one can construct $\mathcal{D}_\tau'$ and apply standard offline RL algorithms to train a high-level policy $\pi(z_i' \mid s_{i \cdot T'})$, which can be used for $T'$-interval control.

## G. Limitations and Future Directions

In this work, we assume that the preference-aligned behaviors learned from crowd preference data are sufficiently expressive to solve the downstream task and share the same implicit objectives with it. However, this assumption may not always hold. One potential direction to address this limitation is to expand the diversity of annotation rewards by incorporating feedback from a larger set of users, while explicitly selecting crowd preference data whose goals are aligned with those of the downstream task. This could ensure that the distribution of preference-aligned trajectories is sufficiently diverse and better covers the optimal trajectory distribution for the downstream task.

In addition, while we focus on state–action safety constraints as the shared objective encoded in crowd preferences, real-world scenarios may involve more complex and harder-to-specify shared objectives. An important extension is to consider cumulative cost constraints or, more generally, arbitrary shared objectives. Understanding how skill composition influences broader classes of shared objectives remains an open question; we provide a heuristic discussion in Appendix F. Another challenge lies in evaluating performance on the shared objective. Although we report cost as a proxy for the hidden objective, in real-world scenarios where the implicit objective is unknown, assessing performance becomes nontrivial. One potential direction is to solicit additional human preferences to evaluate performance with respect to these implicit objectives.

Furthermore, in practical settings, crowd preferences may involve a larger and more diverse set of users. Notably, our method captures latent task rewards rather than user identities, so its scalability depends primarily on the number of distinct preference modes rather than the number of users, which is often much smaller in practice. However, when the number of distinct latent rewards becomes large, performance may be limited by model capacity and potential mode collapse, suggesting an important direction for future work (e.g., exploring mixture or hierarchical latent models). In addition, as the number of users increases, it becomes more likely that some users exhibit inconsistent safety criteria (e.g., violating constraints). These challenges further motivate improving the robustness and scalability of the proposed method as an important direction for future work. Finally, our experiments rely on synthetically annotated preferences generated from ground-truth reward functions, rather than real human feedback. Whether the shared safety structure assumed in our framework faithfully emerges from naturalistic human annotations—which may be noisier, more subjective, and less consistent—remains an open empirical question and an important direction for future validation.

