# OpenReview forum: "Implicit Safety Alignment from Crowd Preferences"
_ICML.cc/2026/Conference — ICML 2026 regular_

### Official Review · Reviewer_7KGH · 2026-03-09

**Soundness:** 3
**Presentation:** 4
**Significance:** 3
**Originality:** 3
**Overall Recommendation:** 4
**Confidence:** 4

**Summary:**

This paper studies how to extract implicit safety criteria from crowd preference data — where multiple users with diverse task objectives share common safety constraints — and transfer them to downstream RL tasks that only have safety-agnostic rewards. The authors first analyze theoretically why naively combining a reward model learned from crowd preferences with the task reward is problematic: the learned reward can be biased by dominant annotators (Theorem 4.3) and requires sensitive weight tuning. They then propose Safe Crowd Preference-based RL, a hierarchical framework that learns latent-conditioned low-level skills from crowd preferences (via VPL or a novel CPL variant) and composes them with a high-level policy that optimizes only the downstream task reward. By constraining actions to the skill space, safety is implicitly inherited. Experiments on six safe RL environments show the method substantially reduces safety violations compared to task-only baselines while matching oracle performance that has access to ground-truth safety signals.

**Compliance With Llm Reviewing Policy:**

Affirmed.

**Final Justification:**

The authors have fully addressed my questions and comments. I will keep my current score.

**Key Questions For Authors:**

1. Suboptimal low-level policies: How sensitive is the safety guarantee to the quality of low-level skill learning? Can you provide empirical analysis (e.g., intentionally degrading skill quality) showing at what point safety violations start appearing? This would help calibrate the practical reliability of the theoretical result in Corollary A.6.
2. Scalability of the skill space: The latent dimension is fixed at 4 (Table 2). How does performance change as the number of distinct crowd preferences grows? Is there a point where the VAE's capacity becomes a bottleneck, and how would you diagnose this in practice?
3. Comparison to Safe RLHF baselines: The paper positions itself against vanilla reward combination but does not compare to Safe RLHF methods (Dai et al., 2023) that jointly learn reward and cost models from preferences, even though these are discussed in related work. While the authors argue their setting doesn't assume separate preference labels for safety vs. task, an ablation comparing against Safe RLHF with synthetically separated labels would help contextualize the gains from the proposed approach.

**Limitations:**

The authors provide a reasonable discussion of limitations in Appendix F and the Impact Statement. I would additionally flag the strong assumption of universal safety consensus as limitations that should be discussed more prominently in the main text rather than only in appendices.

**Strengths And Weaknesses:**

### Strengths

- The problem formulation addresses a genuine gap between Safe RLHF and standard RLHF. Extracting transferable safety criteria from crowd preferences without explicit safety labels is well-scoped, and the paper builds a clear narrative from the failure analysis of reward combination to the proposed skill composition approach.
- Theorem 4.3 is the strongest theoretical contribution. It formalizes how preference imbalance causes the learned reward to collapse onto the dominant annotator's utility, provides a concrete threshold condition on p(zk), and is validated empirically in Figures 1 and 3.
- Restricting downstream optimization to the skill space is a well-motivated design choice. It converts an ill-posed reward decomposition problem into tractable constrained optimization, sidestepping the disentanglement issue.
- Experimental coverage is adequate. Six environments, online and offline settings, balanced and imbalanced regimes, and ablations over regularization, noise, and dataset size. The imbalance experiments (Figure 2, Table 9) are the most informative as they directly test the claimed advantage over reward combination.

### Weaknesses
- The safety guarantee (Corollary A.6) assumes each low-level policy is optimal w.r.t. its conditioned utility, which never holds in practice. The paper does not analyze what happens under suboptimal skills — the high-level policy could compose imperfect skills into unsafe state-action pairs not encountered during training. The regularization term partially mitigates this, but the gap between theory and practice is not adequately addressed.
- The shared safety model (Eq. 4) is restrictive: binary state-action indicators with K→∞. The theory and experiments only cover hard constraint violations. Appendix E honestly acknowledges that cumulative cost constraints break under per-step skill switching, which suggests the safety properties are more fragile than the main text conveys. This can limit real-world applicability.
- Safe-CPL underperforms Safe-VPL on several environments (e.g., Ant-vel: 0.74 vs 0.91 normalized reward) but the two variants are presented symmetrically. A more candid discussion of when each variant is preferable, and why CPL is less stable, would be useful.
- No comparison to Safe RLHF baselines (Dai et al., 2023) that jointly learn reward and cost from preferences. While the authors argue their setting doesn't assume separate safety labels, an ablation with synthetically separated labels would help contextualize the contribution against the most relevant prior work.

---

> ### Author Rebuttal · Authors · 2026-03-29
>
> We thank the reviewer for the feedback and address each point below; we will incorporate them in the revision.
> ## W1. Suboptimality & Safety
> Please see the response to Reviewer unbT’s W1 where we establish the relationship between skill suboptimality (measured by the utility gap to the real optimal policy) and downstream safety violation bound.
> ## W2. Hard Constraint Assumption
> Our work primarily focuses on hard constraint violations because it provides a clear and verifiable definition of crowd preference consensus, which we believe is important given the limited prior work and inherent difficulty in defining shared structure. Moreover, many practical safety requirements can be naturally formulated as hard constraints, which also motivates our settings.
>
> Regarding non-large K and general constraints, we refer to our response to Reviewer X3uH’s W1, where we discuss how safety consensus and transfer may extend. Moreover, as discussed in App E, per-step skill switching is sufficient for hard constraint, while longer intervals help maintain safety for more general forms of constraint but with potential cost of task performance. Such safety-task trade-off can be an important direction for future work.
> ## W3. CPL vs VPL
> As noted in Sec. 5.2, VPL requires additional offline data during skill discovery, whereas the CPL variant relies only on the preference dataset itself, making it preferable under limited data. The two variants also suit different preference model assumptions (partial-return-based vs. regret-based).
>
> Regarding performance, CPL is comparable to VPL in most tasks except Ant-vel. We identify the likely cause as a data collection mismatch: in our implementation, trajectory pairs and user preferences are sampled independently, so the preference used to generate a trajectory may differ from the one used to label it. This misalignment is especially pronounced in Ant, where different preferences correspond to significantly different optimal trajectories. Since CPL resembles preference-weighted behavior cloning, it degrades when high-return trajectories aligned with the annotated preference are scarce. As evidence, enforcing consistency between trajectory-generating and annotating preferences substantially improves CPL performance.
> ## W4. Safe RLHF Baseline
> Thanks for the suggestion—we agree this will strengthen our paper.
>
> Following Safe RLHF, we add an $r_{safety}$-only baseline, where preference data is generated using only safety rewards, allowing a unimodal model to capture safety alone. Results (https://imgur.com/a/v41Z1KI) show its performance comparable to the Oracle with ground-truth safety rewards, as they both directly access a separated safety signal. In contrast, our method—without separated preferences or ground-truth rewards as safety signals—achieves similar safety (within 2% of the Task-only baseline cost) with only a small task performance drop.
>
> Compared to new baseline, RC with a similar pipeline trained on coupled preferences performs much worse, confirming that our mixed setting is more challenging than Safe RLHF settings. Our approach can be viewed as an implicit approximation to Safe RLHF but in this harder setting, where safety cannot be cleanly separated into a reward model and is instead enforced through the composition of safe skills.
>
> We will include these results in the revision.
> ## Q1. Sensitivity to Skill Quality
> Ablations on noise ratio and data size (Fig. 4, 9) provide indirect analysis: increasing label noise or reducing dataset size gradually degrades skill quality by corrupting the safety signal, degrading cost performance while task performance remains stable.
>
> We also add an ablation on low-level policy training steps to explicitly vary skill quality (https://imgur.com/a/rMm1mME). Notably, even with 1e3 steps, safety remains strong while task performance degrades, suggesting safety consensus is more readily preserved even under limited skill training.
> ## Q2. VAE Capacity Bottleneck
> We observe that preference embeddings from different users exhibit significant differences only along 1 latent dimension (for 2 users) and 2 dimensions (for 4 users), even when the dimension is set to be large. This observation motivates our choice of a 4-dim latent space for the main experiments with up to 4 users.
>
> We add an experiment with larger crowd sizes by sampling more target velocities/directions as distinct user preferences. Results (https://imgur.com/a/HLXhEYK) show that increasing the number of users leads to a modest performance drop, while maintaining much better safety than the task-only (unsafe) baseline, even with 16 users in Circle and 32 users in Swimmer.
>
> To diagnose capacity bottleneck in practice, one can monitor reconstruction error and downstream performance. Per-dimension posterior variance is also informative: we observe, as user count grows, more dimensions exhibit large variance, reflecting increased latent capacity utilization.
> ## Q3.
> See the response to W4.

---

> > ### Author Rebuttal · Reviewer_7KGH · 2026-03-31
> >
> > I would like to thank the authors for the clarifications and additional experiment results. Most of my concerns are addressed.

---

> > > ### Author Response · Authors · 2026-04-01
> > >
> > > We are pleased that our revisions and additional experiments have addressed the reviewer’s concerns. We sincerely thank the reviewer again for the insightful comments, which have meaningfully guided the improvement of this paper.

---

### Official Review · Reviewer_X3uH · 2026-03-12

**Soundness:** 2
**Presentation:** 2
**Significance:** 2
**Originality:** 2
**Overall Recommendation:** 4
**Confidence:** 4

**Summary:**

This. paper focus on safety implicitly included in crowd preference datasets.  The proposed method, Safe Crowd Preference-based Reinforcement Learning, contains two stages. The first stage to model diverse user-preferred behaviors as low-level skills and next stage to train a meta composer to control the low-level skill via maximizing the task-specific reward.

**Compliance With Llm Reviewing Policy:**

Affirmed.

**Final Justification:**

The rebuttal addressed my main concerns adequately. The generalized penalty formulation (W1) is a reasonable extension, though formal proofs were deferred to the revision. New results on the VAE-based reward baseline (Q2) support the claim that improvement stems from policy composition rather than multi-modal modeling. The explanation for CPL instability (Q3) is plausible.

Two issues remain partially open: experimental environments share a narrow safety pattern (binary constraints only), and the promised generalized analysis is not yet available for evaluation. Still, the core idea of composing skills rather than combining rewards is well motivated and the experiments are systematic within their scope. I maintain my score of 4 (Weak Accept).

**Key Questions For Authors:**

Q1: When the safety penalty K is not too high in relation to r_user, which indicates that Theorem 4.2's condition is not satisfied, what happens to the method? In this simple case, safe-unsafe trajectory pairs might not be consistent across all users, and the safety signal in crowd preferences might be unclear.

Q2: The present approach uses a VAE-based latent-conditioned model, while the reward combination baseline uses a single unimodal reward model. Have the authors considered creating a baseline that combines rewards using the same VAE-based reward model?

Q3: The CPL variant requires a two-stage training procedure to prevent instability in joint encoder-policy training. I'm curious about more detial on the nature of this instability? Are there issues with the reward scale, conflicting gradient directions, or the posterior collapse of the VAE?

**Limitations:**

yes

**Strengths And Weaknesses:**

Strengths:
This paper attempts to address an interesting problem. In real-world settings, reward design usually does not encode safety requirements. Defining safety transfer as a hierarchical problem is an interesting conceptual contribution.


Weaknesses:
This work does not consider some scenarios.  Such as "how the proposed method perform when safety is not cleanly separable from user-specific preferences" or "when the safety penalty has varying magnitudes rather than a single large constant." Additionally, only a small variety of safety-task interaction patterns are seen in experimental tasks. Tasks have the same type of obvejctive but with a binary constraint item (region or velocity limits)

---

> ### Author Rebuttal · Authors · 2026-03-29
>
> We thank the reviewer for the feedback. We address each point below and will incorporate relevant changes in the revision.
>
> ## W1. Safety-Preference Separability & Varying Penalty Magnitude
> When penalties vary across states, those with sufficiently large penalties (e.g., exceeding Theorem 1’s threshold) are consistently dispreferred by all users, so our safety guarantees still holds on avoiding these highly unsafe states. More generally, if safety is not cleanly separable from user-specific preferences but users still share some common safety structure (reflected in preferences as a shared aversion to certain undesirable trajectories), our method can still catch this shared component through crowd preferences, and the downstream policy remains safe with respect to this shared aspect.
>
> Moreover, the **non-binary, non-large-penalty** setting can be naturally and equivalently subsumed into a more general framework where users follow constraints on a shared cost function $c(s,a) $ with different thresholds. In this case, all users disprefer trajectories that violate the most permissive constraint (i.e., the largest threshold), which forms a new shared safety consensus. For example, if user A has threshold $a>=0$ and user B has $b>=0$, then all users prefer trajectories $\tau$ with $C(\tau)<=\max(a,b)$ over $\tau'$ with $C(\tau') >\max(a,b)$. As discussed in Appendix E, this consensus is more likely to be preserved with longer switching intervals.
>
> Finally, we believe that it's meaningful and reasonable to first focus on the hard constraint assumption as a starting point for studying safety consensus since this choice provides a clear and verifiable definition of crowd preference consensus, which is important given the limited prior work and inherent difficulty in defining shared structure. Moreover, many practical safety requirements can be naturally formulated as hard constraints, which motivates our settings.
>
> We will include formal statements and proofs for the generalized safety formulation described above in the revised version.
>
> ## W2. Limited Task Diversity
>
> We use environments with target velocity, goal, and direction as user-specific preferences, and region and velocity constraints as implicit safety objectives because: 1) they cover most tasks in Safe RL benchmarks; 2) there is a lack of more complex benchmarks that jointly provide multiple objectives (for preference diversity) and constraints (for shared objectives); and 3) strong task performance of existing methods in our tasks allows evaluation to focus on safety, whereas in harder tasks—where all methods struggle—it becomes difficult to assess the impact of safety gains on actual task competition, due to the lack of a reliable reference.
>
> To broaden evaluation coverage, we also include a preliminary LLM experiment. (see response to Reviewer uMdR’s W1).
>
> ## Q1. When K is not too high
> Please see response to W1.
> ## Q2. VAE-Based Reward Baseline
> Thanks for this suggestion. We agree this will strengthen our paper.
>
> Unlike our method, it is unclear how to dynamically adjust the latent variable $z$ during training for a VAE-based reward. Moreover, when using VAE reward, averaging over $z$ biases the reward toward dominant modes, while sampling $z$ introduces randomness and training instability. We therefore fix the prior mean as $z$ to produce a static reward, replacing the unimodal reward in the RC baseline.
>
> New results (https://imgur.com/a/yjS9EpJ) show that this baseline (“RC w/ VAE reward”) performs similarly to the original RC baseline (“RC w/ Unimodal”). This is consistent with our expectation: both baselines use rewards that entangle shared and user-specific preferences—the core limitation our method addresses—and thus fail to mitigate bias from preference imbalance.
>
> ## Q3. CPL Instability
>
> Empirically, we observe that policy optimization converges much more slowly under the CPL loss than reward model optimization under the BT loss, likely due to the more complex conditional output space. As a result, in the CPL variant where both the policy (as decoder) and the encoder are updated simultaneously, the encoder evolves more rapidly and induces continual representation shifts in the latent conditioning variable, while the policy as decoder adapts more slowly to these changes. This mismatch creates a moving optimization target and hinders stable joint convergence, which may contribute to CPL instability.

---

> > ### Author Rebuttal · Reviewer_X3uH · 2026-04-02
> >
> > I thank the authors for their detailed rebuttal and the effort they have put into clarifying the issues raised. Most of my concerns have been addressed. Therefore, I will maintain my current score.

---

> > > ### Author Response · Authors · 2026-04-03
> > >
> > > We are pleased that our revisions and additional experiments have addressed the reviewer’s concerns. We sincerely thank the reviewer again for the insightful comments, which have meaningfully guided the improvement of this paper.

---

### Official Review · Reviewer_unbT · 2026-03-12

**Soundness:** 3
**Presentation:** 3
**Significance:** 3
**Originality:** 3
**Overall Recommendation:** 5
**Confidence:** 4

**Summary:**

The authors address the problem of discovering and transferring implicit safety criteria from crowd preference data to downstream reinforcement learning tasks. The key observation is that while different users in a crowd may have diverse task-specific objectives, they often share common safety-related behavioral criteria (such as avoiding unsafe states). The authors first analyze a naive reward combination approach to learn a single reward model from crowd preferences and combining it with a downstream task reward. From this, the authors identify two fundamental limitations: (1) the learned reward model captures not only shared safety criteria but also user-specific components, which can degrade downstream performance especially under preference imbalance; and (2) the performance is highly sensitive to the trade-off weight between the learned and task rewards.

The authors propose Safe Crowd Preference-based Reinforcement Learning, a hierarchical framework that composes policies rather than rewards. The method first learns latent-conditioned low-level skills from crowd preference data using a VAE-based architecture combined with either standard reward-based RLHF (VPL) or a novel Contrastive Preference Learning (CPL) variant. A high-level policy then composes these skills to solve downstream tasks by optimizing only the task reward, while the skill space implicitly enforces the shared safety criteria. Theoretical results show that the learned skills preserve safety and that better VAE encoding quality leads to more diverse and effective skill sets. Experiments on six safe RL environments demonstrate substantial safety cost reductions compared to task-only baselines, with task performance comparable to oracle methods that have access to the true safety reward.

**Compliance With Llm Reviewing Policy:**

Affirmed.

**Final Justification:**

The rebuttal by the authors answered my questions. The paper is well written, and the authors seem to understand the next steps to have a paper worhy of the conference.

**Key Questions For Authors:**

1. How sensitive is the method to violations of the binary safety assumption (Eq. 4)? If safety penalties are graded rather than binary, do the theoretical guarantees still hold approximately?
2. How does the method perform when the number of crowd users is large (e.g., hundreds) and their safety criteria are only approximately shared? The current experiments use a small number of distinct user reward functions
3. What is the computational overhead of the two-stage training (skill discovery + downstream composition) compared to the reward combination baseline?

**Limitations:**

Yes

**Strengths And Weaknesses:**

## Soundness
* (**Strength**): The theoretical analysis is generally well-developed. Theorem 4.2 shows that a single reward model learned from crowd preferences will correctly rank safe vs. unsafe trajectories under a sufficient penalty condition, and Theorem 4.3 formalizes the risk of preference imbalance.
* (**Weakness**): Minor note---The safety guarantee in Corollary A.6 relies on the assumption that each low-level policy is optimal with respect to its conditioned utility. In practice, policies are only
approximately optimal due to finite data and function approximation errors. The authors could strengthen their contributions by providing a result on the suboptimality of low-level policies translated to downstream safety violations.


## Presentation
* (**Strength**): The paper is generally well-written with a clear narrative. The motivating examples in Figure 1 do a good job of illustrating the limitations of reward combination under imbalanced preferences
* (**Weakness**): The relationship between the offline and online downstream settings is only clearly delineated in the appendices. The main paper would benefit from a more explicit discussion of when each setting is appropriate. Additionally, some notation is introduced without sufficient context (for example, the distinction between z and z′ is initially confusing)

## Significance
* (**Strength**): The problem of extracting reusable safety criteria from diverse crowd preferences is practically important. Most existing RLHF methods assume a single annotator perspective; addressing crowd heterogeneity while extracting shared safety signals is a meaningful contribution to both the safe RL and preference-based RL communities.
* (**Weakness**): The experimental environments are relatively simple, continuous control tasks (locomotion with velocity constraints). It is unclear how well the approach would scale to higher-dimensional or more complex safety specifications, such as those encountered in autonomous driving or language model alignment, both motivating problems from the introduction. A discussion on the gap between the crowd preference setting studied here and real-world annotation scenarios can strengthen the argument.


## Originality
* (**Strength**): The idea of composing policies (skills) rather than combining rewards to transfer safety criteria is novel and well-motivated.
* (**Weakness**): The hierarchical skill-composition framework draws heavily on existing work in skill discovery and variational preference learning. While the combination is novel, the individual components are well-established and not new.

---

> ### Author Rebuttal · Authors · 2026-03-29
>
> Thank the reviewer for the feedback. We address each point below.
> ## W1. Suboptimality & Safety
> To analyze the impact of suboptimality on safety, we consider a low-level policy $\pi(\cdot|z)$ with a utility gap $\delta_z$ (utility is $\sum_t r_{user}+r_{share}$) relative to the optimal policy. Then, the expected violation number is bounded by $\delta_z / K$, as in the worst case the entire utility gap can be attributed to safety violations. If the downstream policy uses at most $m$ low-level policies $\pi(\cdot|z_j)$ per trajectory, the total number of violations is bounded by $ \frac{1}{K} \sum_{j=1}^{m} \delta_{z_j} \le \frac{m\max_z\delta_z}{K}$. We will add the formal analysis to the paper.
> ## W2. Clarity & Notation
> We will add a clearer discussion in the main paper to delineate offline and online downstream settings, including applicability, and further clarify notation.
> ## W3. Scalability & Real-World Gap
> We use environments with target velocity, goal, and direction as user-specific preferences, and region/velocity constraints as implicit safety objectives because: 1) they cover most tasks in Safe RL benchmarks; 2) there is a lack of more complex benchmarks that jointly provide multiple objectives (for preference diversity) and constraints (for shared objectives); and 3) strong task performance of existing methods in our tasks allows evaluation to focus on safety, whereas in harder tasks—where all methods struggle—it becomes difficult to assess the impact of safety gains on actual task competition, due to the lack of a reliable reference.
>
> We also include a preliminary LLM experiment to broaden evaluation (see Reviewer uMdR’s W1).
>
> Extending our method to real-world scenarios requires addressing additional challenges, which also point to future directions. In practice, crowd preferences may involve 1) larger, more diverse user populations, 2) potentially inconsistent safety criteria (e.g., some users violate constraints), and 3) more complex and harder-to-specify shared objectives. Nevertheless, our method's performance gains on evaluated tasks provide a meaningful step toward more practical settings. We will include this discussion in the revision.
> ## W4. Novelty over Prior Work
> We’d like to clarify that our core contribution is introducing a novel and under-explored problem: discovering and transferring implicit safety criteria shared in crowd preferences, along with an intuitive and empirically effective solution. Moreover, our method includes algorithmic contributions: (1) an CPL-based variant of VPL that relaxes data requirements while achieving comparable performance; and (2) a hierarchical downstream loss with a regularization term to mitigate OOD unsafe behavior, compatible with both offline and online settings.
> ## Q1. Non-Binary Safety
> When penalties vary across states, those with sufficiently large penalties (e.g., exceeding Theorem 1’s threshold) are consistently dispreferred by all users, so our guarantees holds on avoiding these highly unsafe states.
>
> This setting can be equivalently viewed as users following constraints on a shared non-binary cost function with different thresholds. In this case, all users disprefer trajectories violating the most permissive constraint (largest threshold), which can be viewed as a shared safety consensus. As discussed in App E, this consensus is more likely to be preserved with longer switching intervals.
>
> We will include a formal statement and proof in the revision.
> ## Q2. Crowd Size
> We conduct experiments with varying crowd sizes by sampling target velocities or directions at regular intervals to construct user-specific preferences. Results (https://imgur.com/a/HLXhEYK) show that scaling to 16 users (Circle) and 32 (Swimmer) yields only modest performance drop while maintaining much better safety than the task-only (unsafe) baseline.
>
> Moreover, our method captures latent task rewards rather than user identities, so its scalability is limited by the number of distinct preferences. In practice, while users may be many, preference diversity is often much smaller, under which our method remains effective. However, if distinct latent rewards become numerous, performance may be limited by the VAE capacity and mode collapse, which we view as an important direction for future work (e.g., exploring mixture or hierarchical latent models).
>
> For approximately shared safety criteria: (1) if users differ in thresholds, see Q1.; (2) if the shared constraints are partially violated, we expect overall safety performance to degrade, as shown in flipping prob experiment in Sec. 6.3.
> ## Q3. Overhead
> The overhead comes from upstream skill discovery (VAE reward + low-level policies), adding up to 4h per task on a GPU compared to the RC baseline (single reward learning), while downstream cost is comparable (~1h, single GPU). Yet, as shown in Fig. 7 (App D.1), the learned skills enable faster convergence than training from scratch, potentially reducing training time.

---

> > ### Author Rebuttal · Reviewer_unbT · 2026-04-04
> >
> > Thank you for your detailed responses to my questions. The authors have satisfactorily answered my questions, so I have updated my score accordingly. Good Job!

---

> > > ### Author Response · Authors · 2026-04-04
> > >
> > > We are truly delighted that our responses have addressed your concerns and sincerely thank you for updating your score. We greatly appreciate your positive feedback and kind words.

---

### Official Review · Reviewer_uMdR · 2026-03-12

**Soundness:** 3
**Presentation:** 3
**Significance:** 2
**Originality:** 2
**Overall Recommendation:** 4
**Confidence:** 3

**Summary:**

Summary:
- The proposed method works as follows:
    - Suppose we have access to a preference dataset of agent trajectories.
    - The preference dataset contains pairs of trajectories, where one trajectory is preferred and the other is not preferred. In this setting there is no access to the rewards.
    - The proposed method trains a VAE that learns latent encodings z’ for crowd behaviors, as well as a reward function.
    - They then train skill policies given the latent and reward functions.
    - For the specific task reward, they train a high level policy \pi_h that generates the high level z’.
- For evaluation, the paper demonstrates the success of the method on Bullet-Safety and Safety-Gymnasium environments.

**Compliance With Llm Reviewing Policy:**

Affirmed.

**Final Justification:**

The authors have sufficiently addressed my main concerns. I originally asked questions regarding the evaluation settings, as well as various clarifications regarding plots, differences with prior work, etc. The authors have addressed these in the rebuttal with some limited evaluations in language domains, as well as clarifications to my questions regarding the plot in Figure 1, differences with OPAL, etc. I'm inclined towards a weak accept, though I would have preferred to see more settings beyond the standard safe RL benchmarks.

**Key Questions For Authors:**

Questions:
- Could the authors clarify Figure 1? I’m having trouble understanding the plot. What are major and minor pref in these plots? It also appears that safe samples have only slightly higher reward for the Half-Cheetah-vel environment. Am I understanding this correctly?
- Can the authors clarify differences with this paper (https://arxiv.org/pdf/2010.13611), titled “OPAL: Offline Primitive Discovery for Accelerating Offline Reinforcement Learning”? I’m struggling to see how different this policy composition really is. The only difference I see is the replacement of the reward function with a preference model.
- Is it necessary to frame the approach as “Safe Crowd Preference-Based RL”? From what I can tell, this same framework can be applied in preference-related settings outside of the context of safety.
- How is the “safe SAC agent” in the “Dataset” subsection of the Experiments trained? From what I can tell, this would be using a balance between the goal-specific and safety rewards — is that right?
- Given the paper framing around crowd preferences, have the authors considered evaluating their method in LLM/agentic use cases? I think it’s clear that this method works reasonably well in standard safe RL benchmarks.

**Limitations:**

None in particular, other than the need to evaluate the method on LLM/agentic environments, given the focus on crowd preferences. See the weaknesses and questions.

**Strengths And Weaknesses:**

Strengths:
- This paper provides a theoretically motivated approach for learning from crowd preference data.
- The paper avoids key RLHF challenges, including the weighting problem and the extraction of safety criteria, by implicitly discovering this through a policy composition, inspired by hierarchical RL techniques.
- The paper’s approach to learning from crowd preference data is realistic, as it is often the case that rewards from crowd preference data are not available.
- The paper evaluates the proposed methods in standard safe RL environments (i.e., Gym-style environments) and demonstrates that their method is able to strike a Pareto-frontier balance between task success and safety.

Weaknesses:
- The paper focus is on crowd preference data, though the studied environments are in safe RL “Gym”-style environments. Crowd preference is a term closely associated with human behavior, often exhibited through natural language. This work is missing evaluation in natural language domains.
- It is unclear what happens if a new task requires behaviors that may be unsafe. In this work, the authors assume a fixed set of possible unsafe behaviors, as encoded with the VAE, though one could imagine a new task that may uncover new unsafe behaviors that are not accounted for in the safety preference reward model. The authors should comment on how the method would adapt to new tasks with completely different safety constraints.
- While the method avoids the weight hyperparameter w for balancing between the r_new and r_shared, the method still has a hyperparameter \beta_{reg} to tune. The paper does consider an ablation in this regard, though it is worth pointing out the need for hyperparameter tuning in this method.

---

> ### Author Rebuttal · Authors · 2026-03-28
>
> Thank the reviewer for the feedback. We address each point below and will incorporate relevant changes in the revision.
> ## W1. LLM Evaluation
> Crowd preference is an important problem beyond LLM settings, including domains like robotics where crowd datasets are common [1–2]. Prior work on crowd preference learning has also evaluated on non-LLM domains [3–5], motivating our evaluation.
>
> Here, we additionally conduct a preliminary LLM evaluation. Following VPL [5], we construct a synthetic LLM dataset with four response types (A, B, C, D). Preferences are from two users with rankings A>B>C>D and C>B>A>D, where D can be considered unsafe as it is dispreferred by all users.
> In the downstream task, each state is a pair $s=(y,y')$, and the agent selects between them. It receives $r_{task}=1$ only if the choice matches a new downstream preference (e.g., D>B>A>C), which conflicts with crowd safety. Our method learns a VAE reward and trains $\pi_h(z|s)$ to select rewards used for decision-making. We compare with (1) Task-only, which directly optimizes task reward; and (2) RC($\omega$), which combines $r_{task}$ and unimodal RLHF reward. We evaluate on two subsets of states—those with D and without D—and report task reward $R$ and safety violations $C = N(\text{choice D})$.
>
> Results (https://imgur.com/a/6Ibl901) show: Task-only always selects D in With-D states, causing violations. Our method achieves optimal task performance on No-D and avoids unsafe choices on With-D. RC either violates safety or sacrifices task performance due to preference imbalance bias.
>
> ## W2. Downstream Safety
> While downstream tasks incentivize unsafe behavior, our method avoid it by restricting optimization to the preference-aligned (safe) behavior set, which may slightly reduce task performance but remains close to an oracle baseline with access to true safety rewards. Empirically, many evaluated tasks already incentivize unsafe behavior (e.g., X-vel tasks encouraging faster movement despite velocity constraints), yet our method maintains strong safety.
>
> If a downstream task introduces entirely new constraints not reflected in the crowd preferences, the method may be limited, as no corresponding signal is available.
>
> ## W3. $\beta_{reg}$ Tuning
> As shown in Fig. 4(a) and 8, $\beta_{reg}$ is more robust than the weight $\omega$ between $r_{new}$ and $r_{shared}$. Across a wide range of $\beta_{reg}$, it has only a minor impact on task reward while consistently yielding substantial safety improvements, so tuning $\beta_{reg}$ in practice should impose only a small burden. We will clarify this in the revision.
> ## Q1. Clarification of Figure 1
> All users prioritize safe over unsafe trajectories. Among safe trajectories, the majority in Run prefers velocity 0 (blue line), while a minority prefers velocity 5 (yellow); in HalfCheetah, the majority prefers higher positive velocity, while a minority prefers higher negative velocity. Rewards are higher for majority-aligned trajectories than minority ones, consistent with the crowd imbalance bias we discuss.
>
> Across both tasks, safe trajectories (green line) have higher expected reward than unsafe ones (red), providing a sufficient signal to distinguish safe from unsafe behavior. However, the reward gap may not reflect the true reward difference, as its magnitude is often sensitive to data and reward training process.
> ## Q2. Difference from OPAL
> For policy composition, OPAL encodes trajectory segments into transitions with latent z as actions and trains policies on them, while we essentially train $\pi_h$ over raw transitions with $\pi_l$ as an additional policy layer. We further incorporate OOD skill regularization and support both offline and online settings. Motivations also differ fundamentally: improving offline RL vs. uncovering shared safety structure.
> ## Q3. Framing as “Safe Crowd Preference-Based RL
> We believe this framing is appropriate, as our work focuses on settings where safety constraints serve as a shared objective across crowd, enabling a clear notion of consensus and evaluation. We agree that the framework could be extended to more general forms of crowd consensus, but consensus beyond safety is harder to define and verify, and its transfer via policy composition may not be theoretically guaranteed.
>
> ## Q4. Safe SAC Agent Training
>
> Yes. it is trained with a fixed weight between user-specific and safety rewards. We find it's sufficient to achieve near-zero cost while maintaining sufficient exploration for data collection. See App C.2 for more details.
>
> ## Q5. See response to W1.
> [1] The Impact of VR and 2D Interfaces on Human Feedback in Preference-Based Robot Learning
>
> [2] Humanoid Everyday: A Comprehensive Robotic Dataset for Open-World Humanoid Manipulation
>
> [3] Crowd-PrefRL: Preference-Based Reward Learning from Crowds.
>
> [4] Pareto-Optimal Learning from Preferences with Hidden Context.
>
> [5] Personalizing Reinforcement Learning from Human Feedback with Variational Preference Learning

---

> > ### Author Rebuttal · Reviewer_uMdR · 2026-04-03
> >
> > Thank you for the responses. Though I would have preferred to see more evaluation in LLM domains, I am inclined towards a weak accept, given that the authors addressed my concerns.

---

> > > ### Author Response · Authors · 2026-04-03
> > >
> > > We are pleased that our revisions and additional experiments have addressed your concerns, and we sincerely appreciate your decision to increase the score.
> > >
> > > We agree that our work would benefit from more extensive evaluation in LLM domains. Due to the limited rebuttal timeframe and the lack of well-established datasets (ideally those capturing diverse preferences under common safety criteria), we began with simpler environments to provide an initial validation. We acknowledge this as an important direction and will continue to expand our evaluation in more diverse LLM settings.

---

### Decision · Program_Chairs · 2026-04-30

**Decision:**

Accept (regular)

**Comment:**

The paper considers the problem of learning safe policies from crowd preferences when user preferences encode distinct objectives but follow common safety principles. The main contribution is a novel hierarchical framework that learns safety-aligned skills from crowd preferences and composes them into a high-level policy to solve downstream tasks. The paper shows why vanilla RLFH is inadequate when the preference dataset is not balanced, and empirically demonstrates the utility of the proposed approach using an experimental testbed based on six RL environments. Overall, the reviewers were positive about this submission, indicating that the paper is generally well-written, the proposed approach well-motivated, and the experimental coverage adequate. The reviewers also mentioned several weaknesses, including the scalability of the approach, the gap between the studied setting and real-world scenarios, and the restrictiveness of some of the assumptions. The rebuttal has addressed most of the reviewers' concerns, and the authors are encouraged to revise their manuscript accordingly.